# Femtosecond laser writing of ant-inspired reconfigurable microbot collectives

Zhongguo Ren[1], Chen Xin [1,2] ✉, Kaiwen Liang[1], Heming Wang[1], Dawei Wang[1], Liqun Xu[1], Yanlei Hu [1], Jiawen Li [1], Jiaru Chu[1] & Dong Wu [1] ✉

Microbot collectives can cooperate to accomplish complex tasks that are difficult for a single individual. However, various force-induced microbot collectives maintained by weak magnetic, light, and electric fields still face challenges such as unstable connections, the need for a continuous external stimuli source, and imprecise individual control. Here, we construct magnetic and light-driven ant microbot collectives capable of reconfiguring multiple assembled architectures with robustness. This methodology utilizes a flexible two-photon polymerization strategy to fabricate microbots consisting of magnetic photoresist, hydrogel, and metal nanoparticles. Under the cooperation of magnetic and light fields, the microbots can reversibly and selectively assemble (e.g., 90° assembly and 180° assembly) into various morphologies. Moreover, we demonstrate the ability of assembled microbots to cross a one-body-length gap and their adaptive capability to move through a constriction and transport microcargo. Our strategy will broaden the abilities of clustered microbots, including gap traversal, micro-object manipulation, and drug delivery.

Collective behavior of organisms in a variety of environments is widespread, and they are able to self-organize in an orderly manner to perform tasks that would be difficult for a single individual to accomplish. Currently, the formation of biological collectives can be divided into two main categories. In the first category there is no direct contact between individuals, as in the case of bees[1], fishes[2], and birds[3], causing these collectives to be unstable and vulnerable to disruption. In contrast, more stable collectives are formed by individuals intertwining with each other through mouth or limb deformations as observed in ants[4,5], or by one biting on the base of the tail of another in the case of shrews[6]. These collectives are formed by individuals locking onto each other. They can resist interference and always remain as an integrated whole.

Inspired by the first collective type, researchers have developed collectives of microbots capable of reconfiguring their morphology in response to external magnetic field[7,8], light[9,10], electric field[11], and ultrasound[12,13], enabling a diversity of assembly patterns. Although these assembly strategies can reconfigure the morphology of micro/nano swarms to perform certain collective behaviors[14–16], the monomers are poorly controllable and cannot be selectively and individually actuated. In addition, the assembled morphology requires continuous input of external stimuli to maintain[10,17]. These drawbacks limit the behavior and functionality of microbot collectives. Therefore, proposing a stabilized assembly strategy similar to that of the second biological collective is crucial for the development of advanced applications of microbot collectives.

Ants, the second collective type, can utilize mouth or limb deformations for interlocking to accomplish stable connection and controllable separation between individuals[18]. Inspired by this behavior, using a deformable microbot as the basic unit is a promising method to construct microbot collectives that can be assembled and disassembled in a controlled and stable manner. With the rapid

[1]Key Laboratory of Precision Scientific Instrumentation of Anhui Higher Education Institutes, CAS Key Laboratory of Mechanical Behavior and Design of Materials, Department of Precision Machinery and Precision Instrumentation, University of Science and Technology of China, Hefei 230026, China. [2]Department of Mechanical and Automation Engineering, The Chinese University of Hong Kong, Hong Kong 999077, China. ✉e-mail: xinc@ustc.edu.cn; dongwu@ustc.edu.cn

advancement of micro/nanofabrication technologies[19–23] and material science[24–29], microbots based on stimulus-responsive materials have been developed, which exhibit excellent responsive deformation properties, high environmental adaptability and powerful functionality. As typical examples, Sitti et al. created cubic liquid crystal elastomer voxels with predefined field orientations directions and stitched these voxels into lines, grids, and skeletal structures for programmable thermally-triggered morphology changes in three dimensions[30]. Duan et al. constructed 3D reconfigurable pH-responsive hydrogel micro-building blocks and realized sophisticated 3D-to-3D shape transformation[31]. The aforementioned examples have demonstrated the superior properties of the proposed stimuli-responsive material microbots. Nevertheless, from the view point of the number of microbots, current researches generally focus on a single deformable microbot. Despite the fact that reversible, dynamic and fast deformation or motion of individual microbot can be realized controllably, stable and reversible connections between multiple deformable microbots to form collectives as well as their locomotion and applications have not been validated.

In this work, inspired by the behavior of ant colonies[32], we construct collectives of ant microbots that are capable of being precisely assembled, disassembled and reconfigured (Fig. 1a). The rigid body and flexible joints of each ant microbot are fabricated by sequential polymerization of magnetic photoresist and thermal stimuli-responsive hydrogel by two-photon polymerization (TPP) process[33–37], and then silver nanoparticles (Ag NPs) are deposited onto the head of the ant microbot based on photoreduction[38,39]. Due to the powerful photo-thermal conversion effect of the Ag NPs, the ant mandibles open immediately when the laser is focused on the ant head. In addition, the ant microbot has an L-shaped tail. According to this design, the microbot collectives can accomplish controllable, reversible and multimodal assemblies by mandible-clamping the tail under the coordinated control of external magnetic and light fields. Ant microbot collectives assembled with mechanical deformation do not require continuous injection of external energy to maintain the assembly morphologies and exhibit strong assembly stability. Furthermore, the strategy realizes not only the assembly of magnetic units, but also the assembly of non-magnetic units with the assistance of a magnetic unit. We demonstrate the ability of assembled ant microbots to traverse a one-body-length gap (Fig. 1b). Finally, as a proof-of-concept, it is displayed that ant microbot collectives are able to switch assembly morphologies to adapt to the environment and accomplish the transportation of microcargos (Fig. 1c). Our strategy may enable microbot collectives with multimodal and multifunctional capabilities, ultimately rendering them powerful tools in the fields of particle manipulation and drug delivery.

## Results

### Design, fabrication and characterization of ant microbot

The schematic of the design and fabrication process of the ant microbot with multi-materials is illustrated in Fig. 2 (Supplementary Fig. 1), and the whole procedure consists of three steps. In the first step, the 3D magnetic photoresist body of an ant is fabricated by TPP process based on a predesigned model (Fig. 2a). Subsequently, after the sample is developed with ethanol and washed with DI water, the thermal stimuli-responsive hydrogel is dropped onto the microstructure. In the secondary TPP process, the two asymmetrical bilayer hydrogel joints are fabricated to connect the mandibles to the body (Fig. 2b). Specifically, the hydrogel joint is composed of a low crosslink density portion and a high crosslink density portion, where the low crosslink density part has a larger shrinkage ratio. SEM images provide a direct comparison of the ant microbot before and after joints integration, and detailed SEM images show the distribution of the low and high crosslink density parts, where their contours are marked in light and dark red, respectively (Fig. 2b). Finally, after the removal of the unpolymerized hydrogel, the silver precursor is added to the microbot, and the Ag NPs coating tightly attached to the surface of the ant microbot head is reduced by two-photon reduction (Fig. 2c and Supplementary Fig. 2), in which the Ag NPs with average roughness of 42.3 nm (Supplementary Fig. 3) are reduced from silver ammonium ions absorbing photons. The differences between before and after reduction of Ag NPs coating are also presented by SEM images, and the contours of Ag NPs are marked in yellow (Fig. 2c). Energy dispersive spectrometer (EDS) images demonstrate the presence of the Fe and Ag elements. In this way, an assembling micro unit incorporating various materials with distinct properties and metal coating can be manufactured (Supplementary Fig. 4).

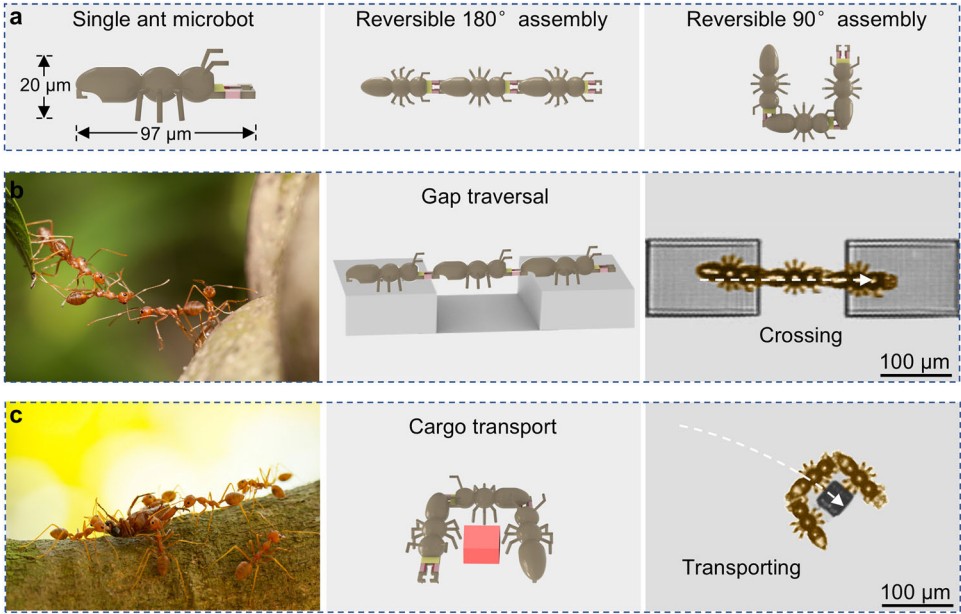

**Fig. 1 | Schematics of ant colony behavior-mimicked reconfigurable three ant microbots. a** The I-shaped assembly and reversible C-shaped assembly process of three ant microbots under magnetic and light fields. **b** Three ant microbots assembled in I shape mimic ants traversing a gap. Image of ants used under license from Depositphotos. **c** Three ant microbots assembled in C shape mimic ants transporting cargo. Image of ants used under license from Depositphotos.

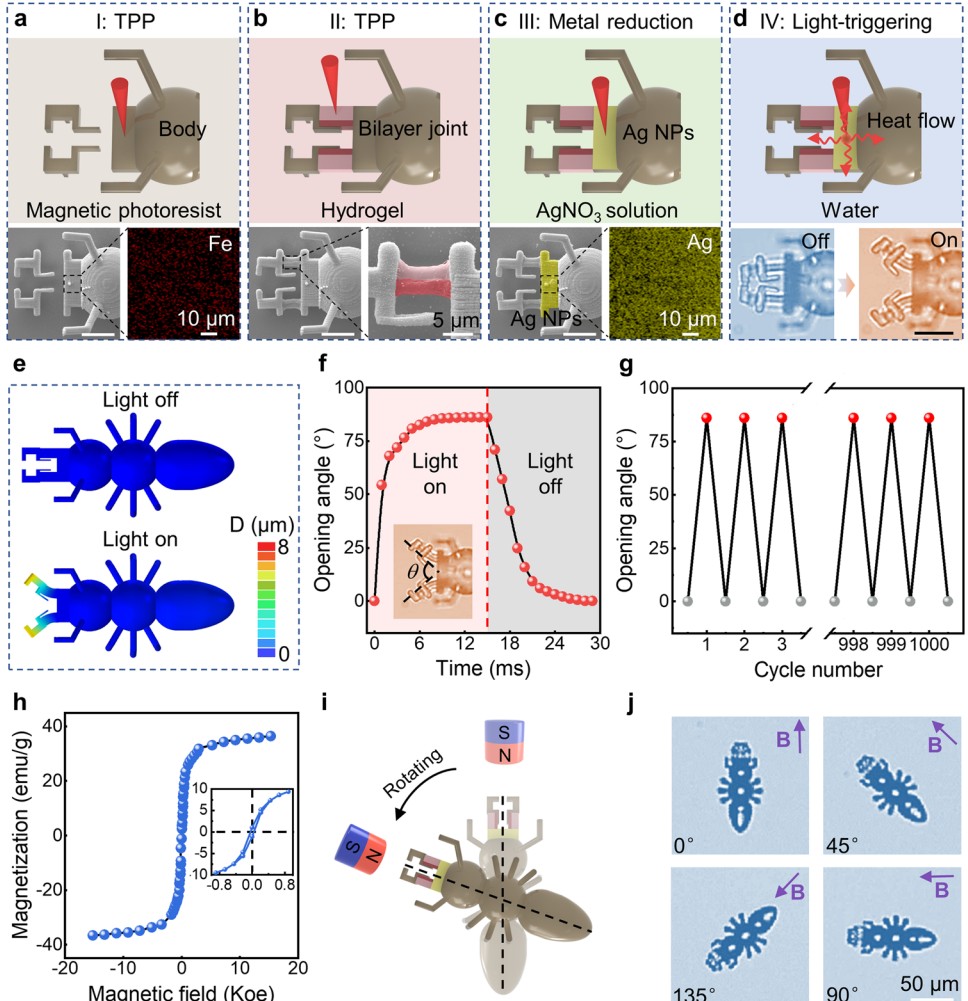

**Fig. 2 | Design, Fabrication and characterization of ant microbot. a** The ant body is fabricated by femtosecond laser direct writing in magnetic photoresist. **b** Thermal stimuli-responsive hydrogel joints are integrated into the body via TPP. The joint consists of two parts with distinct cross-link densities, where dark red and light red portions indicate high and low cross-link density, respectively. **c** Site-selective photo-reduced Ag NPs are used for photothermal conversion, where the Ag NPs are reduced from silver ammonium ions absorbing photons. The area with Ag NPs in the SEM image is marked in yellow. **d** The laser is used to trigger the ant mandibles to open, and the radius of the focused spot under the 5 × objective lens is ~3.3 μm. When Ag NPs are irradiated with NIR light, strong photothermal conversion induces the temperature increase in hydrogel joints. The portion of the asymmetric joint with low cross-link density produces larger shrinkage deformation than the portion with high cross-link density, causing the hydrogel joints to drive the mandibles open. Scale bars, (**a–d**) 10 μm. **e** Simulation results show that the heat converted from light absorbed by the Ag NPs induces the hydrogel joints to deform driving the mandibles to open. **f** Dynamic light-responsive properties of the mandibles, where the response time is ~8 ms. Scale bar, 10 μm. **g** Bending angle of the joints over 1000 cycles of on-off switch of laser with actuation power of 20 mW. **h** The mass magnetization curve of the magnetic ant microbot measured with VSM, which indicates the superparamagnetism of the magnetic ant microbot. **i** If there is an angle between the ant microbot axis and magnetic direction, a magnetic torque is generated on the ant microbot, which induces a rotational motion of the ant microbot to align along the magnetic field. **j** Optical images show the ant microbot rotation with the rotation of an external magnetic field (**B**).

The ant microbot consists of a magnetic photoresist body, an Ag NPs coating and two hydrogel joints. Among them, the hydrogel is mainly composed of N-Isopropyl acylamide (NIPAM) monomer[40,41], Methylene-Bis-Acrylamide (MBA) crosslinker, and Diphenyl (2,4,6-tri-methyl benzoyl) phosphine oxide (TPO) photoinitiator. The deformation mechanism of thermo-responsive hydrogel is thought to be a reversible phase transition from a contracted hydrophobic state at temperatures above its lower critical solution temperature (LCST) to a hydrophilic swollen state at temperatures below its LCST, which is almost between 32 °C and 33 °C[42,43]. Since the mechanical strength of printed NIPAM structures is relatively poor[44], the polyvinylpyrrolidone (PVP, K30) is dissolved in the pre-gel solution to improve the mechanical strength of the hydrogel microstructures (Supplementary Fig. 5), and it has a near-negligible effect on the photothermal sensitivity of the hydrogel[21]. In this work, to quantitatively investigate the expansion and contraction properties of this hydrogel, micro circular plates with a diameter of 20 μm and a thickness of 3 μm are fabricated and measured and full contraction in 8 ms is observed when the microplate is illuminated. We define the shrinking ratio ($\varepsilon$) of the hydrogel structures as the ratio of the length change to the original length, and the processing parameters that affect the shrinkage have been reported in detail in our previous work[21]. In this study, taking advantage of the programmable fabrication capability[45,46], we keep the processing laser power and scanning point step constant at 33 mW and 320 nm, respectively, while choosing scanning repeat time (SRT) as the processing variable to control the shrinking ratio of the hydrogel, and the maximum shrinking ratios for SRT = 1 ms and SRT = 3 ms are ~0.39 and ~0.1, respectively (Supplementary Fig. 6). For hydrogel joints, we systematically investigate the effect of the difference and the width ratio of different crosslink density layers on their bending angle (Supplementary Figs. 7–8). Based on these experimental results, we design a micro asymmetrical double-layered joint (8 μm in length and

4 µm in width) to achieve bending deformation, where the width ratio of the SRT = 1 ms region to the total width is 0.7 and the remaining part of SRT = 3 ms. In the composite ant microbot, the Ag NPs function as the photoreceptor and photothermal agent, which absorb energy from the laser irradiation and then convert it to thermal energy to heat thermo-responsive hydrogel joints. The hydrogel joint consists of two layers with different SRT, which result in distinct degrees of cross-linking, and consequently, the hydrogel joints can exhibit asymmetric deformation. Specifically, as soon as the temperature of the joint is higher than the LCST of the hydrogel via surface incorporated photo absorbers, it immediately bends toward the side of the fewer SRT due to the larger shrinking ratio of the fewer SRT than the more SRT. In this way, the mandibles initially close together, and once the light field perpendicular to the surface of the head is applied, the mandibles will be actuated by the joints to open. When the external illumination is removed, the mandibles return to their original closed condition (Fig. 2d). To obtain insight into the degree of bending deformation caused by different SRTs, theoretical simulation is conducted to verify the bending shape of the joints, and the simulated bending direction and angle match well with the experimental results, indicating the feasibility of this design strategy (Fig. 2e). The dynamic deformation of the joints under 20 mW laser indicates a rapid response of the joints (Fig. 2f and Supplementary Movie 1). It takes only 8 ms to reach a maximum opening angle of ~86°, whereas the recovery to the initial state takes 12 ms after the switch off of the laser. This is understandable because the heating process of hydrogel joints that benefit from high energy density is faster than the cooling process (Supplementary Fig. 9). In addition, the opening/closing behavior of the mandibles is stable and reversible. Figure 2g shows the opening angle of the ant microbot during 1000 times of switching on/off the illumination, and no significant fatigue is observed after repeated stimulation, indicating the stable responsiveness (Supplementary Fig. 10 and Supplementary Movie 2).

The component of the ant microbot body is magnetic photoresist, which is intended to be controlled remotely by introducing an external magnetic field. The magnetic photoresist is prepared by mixing $Fe_3O_4$ magnetic nanoparticles surface-modified with oleic acid with commercial photoresist SZ2080[47]. The vibrating sample magnetometry (VSM) test shows that the remanence and coercivity are all almost zero, indicating the ant microbot is superparamagnetic, which is desirable for movement manipulation (Fig. 2h). A permanent ferromagnet is used for remote control (Supplementary Fig. 11), which applies a magnetic torque and force on the magnetized ant microbot. The magnetic force associated with the magnetic field gradient leads to the translation of the ant microbot, and the magnetic torque induces rotational motion of the ant microbot to be aligned along the magnetic field (Fig. 2i). We test the ant microbot's rotational motion capability by using a Helmholtz coil that can generate a uniform magnetic field (~45 mT), and the direction of the magnetic field is changed by 45° each time. The results show that the response time, standard deviation and overshoot are 13.05 s, ~0.71°, and 2.35°, respectively. (Fig. 2j and Supplementary Fig. 12).

## Reversible, multiple and robust assembly of magnetic ant microbots

The on-demand assembly and separation of the ant microbots are achieved by coordinated control of the magnetic and light fields. As the schematics show in Fig. 3a, there are two ant microbots, and one of them is fixed to the substrate while the other is free to move. When a handheld permanent magnet is carefully approached, the ant microbot in the free state will be aligned along the direction of the magnetic field and will be able to translate under the magnetic force. By adjusting the relative distance and angle between the units as well as leveraging the light field to control the opening and closing of the mandibles, we can controllably accomplish two stable assembly states

by gripping the tail with the mandibles, i.e., tending to 180° assembly when the angle between two units in long axis direction is 180° and tending to 90° assembly when the angle is 90°. Benefiting from the non-contact triggering and structural design, the deformation-based mechanical assembly between units can tolerate large angular and positional errors, and the assembled structure no longer requires a continuous supply of external stimuli to be maintained. On the other hand, benefiting from the hydrogel joints having a stable deformation capability, the assembled units can be easily separated under the combined control of magnetic and light fields. However, since the assembled units are sufficiently stable, they will remain in their original state if they are only subjected to the magnetic field, which is the reason that the new units can be further assembled normally.

To intuitively demonstrate the effectiveness of assembly and configurations of ant microbots, we employ two ant microbots to achieve 180° assembly and 90° assembly sequentially (Supplementary Movie 3-4). As the snapshots shown in Fig. 3b, one ant microbot is anchored to the substrate while the other is randomly distributed. Then, the magnetic field actuates the free unit to approach the fixed unit. When the latter mandibles contact with the former tail, a light field is introduced to drive the latter mandibles open. Afterward, the latter continues to move forward driven by the magnetic field so that the latter mandibles encircle the former tail. Finally, the two ants achieve 180° assembly after turning off the light field. Conversely, we can use magnetic and light fields to disperse the assembled structure into two individual units. Similarly, using this strategy the two ant microbots can also achieve 90° assembly and separate on-demand (Fig. 3b). Notably, the 90° and 180° assembly methods are combinable and extendable to complete multiple configurations. As the demonstrations illustrated in Fig. 3c, by changing the sequence and number of the two assembly modes, numerous configurations of ant microbot can be realized. Here, the 4 units are sequentially assembled into an initial C shape by a 90° assembly, a 180° assembly and a 90° assembly, then the assembly pattern between two neighboring units is adjusted on demand by the control of magnetic and optical fields for further morphological reconfiguration. Benefiting from the controlled and selective assembly, these independent units are reassembled into a J shape. Similarly, with repeated controllable separation and selective assembly, micro units can be reversibly reconfigured into the O, V, S, and W shape by combining two assembly modes. In addition to the ant microbot, our assembly strategy is equally applicable to other shaped microbots with multiple claws and gripping points. Thanks to the flexibility of our fabrication approach, we have also designed other shaped microbots and realized more complex assembly forms (Supplementary Fig. 13). For example, rectangular-shaped microbots with two claws and six gripping points can be constructed into bifurcated tree-like patterns (Supplementary Fig. 14), and disk-shaped microbots with two claws and two gripping points can be assembled into shapes dispersing from the center outward (Supplementary Fig. 15). To manipulate multiple ant microbots at the same time, we can introduce a spatial light modulator to generate multifocal light fields to programmatically control the opening and closing of the mandibles of many ant microbots as well as many claws on a single microbot (Supplementary Fig. 16). In addition, by controlling the concentration of magnetic nanoparticles in the magnetic photoresist, multiple ant microbots can achieve synchronized or asynchronized motions driven by the same magnetic field (Supplementary Figs. 17–18).

The controllable and selective assembly strategy can also enable the assembly between multiple non-magnetic material units with the assistance of a magnetic unit (Supplementary Movie 5-6). Continuous sequence images of this process are shown in Supplementary Fig. 19, a magnetic unit that clamps a non-magnetic unit is guided by a magnetic field close to the other non-magnetic unit. When the mandibles of the second non-magnetic unit contact to the tail of the first non-magnetic unit, a light field is turned on to actuate the mandibles of the second

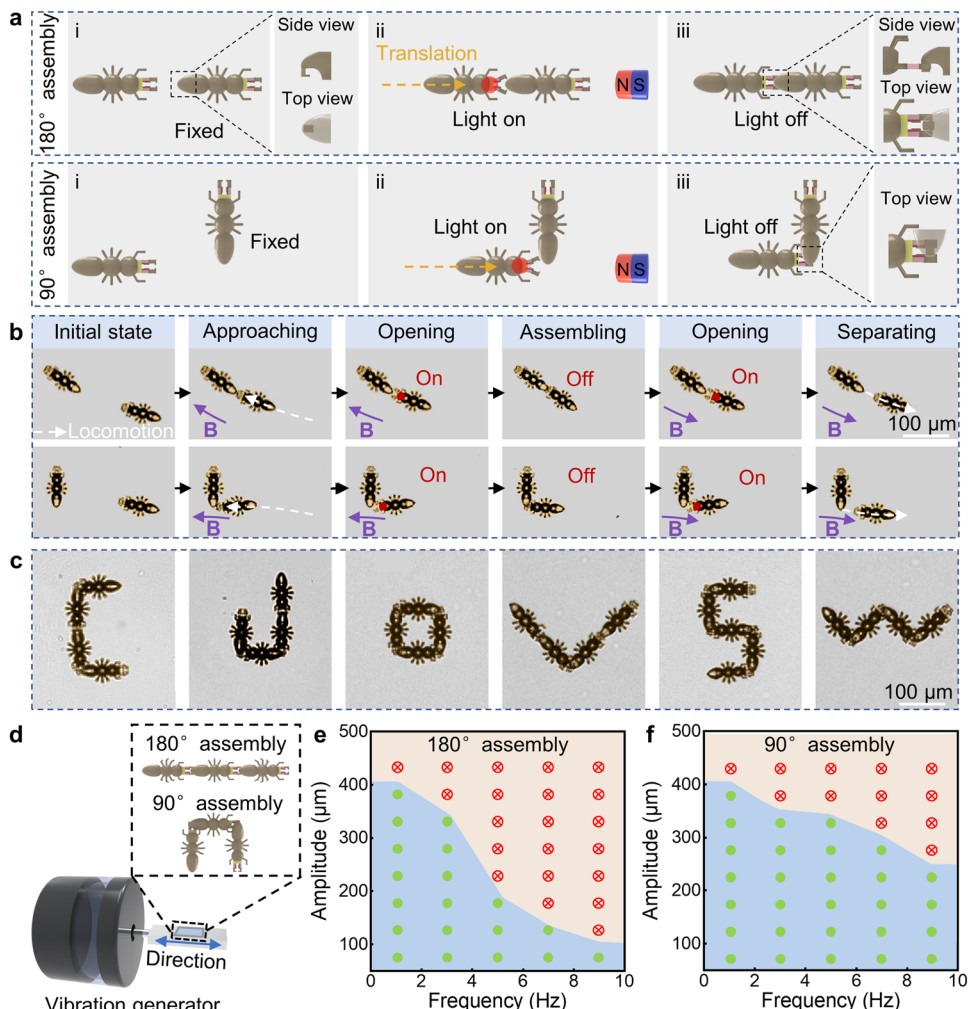

**Fig. 3 | Reversible, multiple, and robust assembly of magnetic ant microbots.** **a** Schematics of the 180° and 90° assembly processes of two ant microbots under the synergistic control of magnetic and light fields, respectively. **b** Time-sequence optical images show the 180° and 90° assembly and corresponding separation processes of two ant microbots under the synergistic control of magnetic and light fields, respectively. **c** Optical images show the versatile morphologies achieved by introducing units' number and combining two assembly methods. **d** Schematic of the vibration test setup. The slide containing ant microbots is mounted on the vibration generator, where the first ant microbot is fixed on the slide and the other two are assembled sequentially by our proposed strategy. **e,f** the phase diagrams reveal I-shaped (**e**) and C-shaped (**f**) assembly stability of three ant microbots under different vibration parameters. The green dots represent that the assembled microstructures remain intact. The red cross symbols indicate that the assembled microstructures are destroyed.

non-magnetic unit to open. After that, the magnetic unit continues to move forward with the non-magnetic unit so that the mandibles of the second non-magnetic unit can encircle the tail of the first non-magnetic unit. Finally, the three units achieve 180° assembly after turning off the light field. In addition, focusing the light spot on the head of the third magnetic unit induces its mandibles to open, which in combination with the control of the magnetic field allows it to separate from the whole, thus realizing the 180° assembly between two non-magnetic units. Similarly, using this method the two non-magnetic units can also achieve 90° assembly and separate on-demand (Supplementary Fig. 19). To illustrate the dynamic control capability of the magnetic unit to manipulate different non-magnetic material units in a single coding sequence, we pattern different material units into various configurations that closely resemble the tetrominoes in the popular game Tetris (Supplementary Fig. 19). Therefore, this controllable and selective assembly strategy provides more possibilities for the assembly of micro unit collectives.

To evaluate the reliability of assembly, we employ a horizontal vibration generator (VG) to investigate the bonding stability between units with 90° and 180° connection respectively. Note that the first unit

in both assembly patterns is anchored to the substrate to vibrate with the VG, while the remaining two units are assembled by our proposed strategy. Figure 3d shows the stabilization of the assembled structure under different combinations of vibration frequency and amplitude (Supplementary Movie 7–8 present the stabilization of two assembled modes at a vibration frequency of 5 Hz with increasing amplitude). The green dots represent that the assembled microstructures remain intact and the red cross symbols indicate that the assembled microstructures are destroyed (Supplementary Fig. 20). It can be seen that the amplitude needed to disperse the assembled architecture gradually decreases as the applied frequency increases for both assembly methods and the stability of the C shape is larger than that of the I shape. Note that the assemble reliability is sufficient to guarantee that the ant microbot collectives can move and stably transport cargo as shown in the following sections.

## Gap traversal with a single ant microbot and assembled ant microbots

In this work, we systematically investigate the locomotion performance of single and assembled microbots steered by a magnetic field

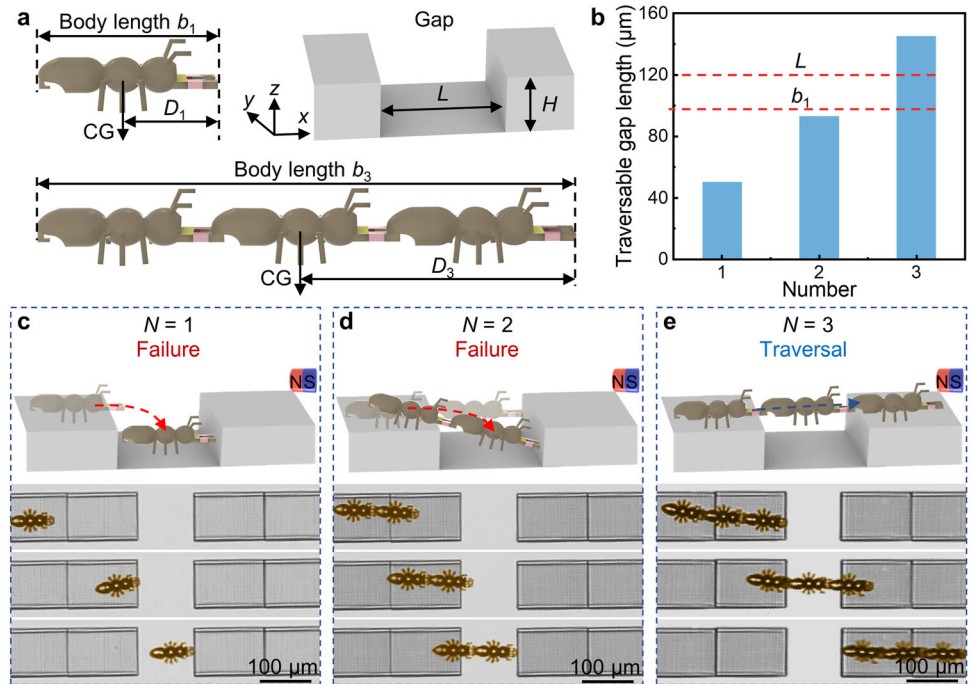

**Fig. 4 | Gap traversal with a single ant microbot and assembled ant microbots.** **a** geometric parameters of the ant microbot and the gap. The body length $b_1$ of a single ant microbot is 97 μm, and the distance $D_1$ between its center of gravity and the front of the mandibles is 50 μm. The length $L$ and height $H$ of the gap are 120 μm and 20 μm, respectively. **b** The length of the gap that can be traversed by the microbot assembled from different numbers of units. **c** Schematic and optical diagrams show that a single microbot fails to cross the gap and falls completely into the gap. **d** Schematic and optical diagrams show that assembled two microbots also fail to cross the gap and become stuck in the gap. **e** Schematic and optical diagrams show that assembled three microbots pass the same gap successfully.

in a gap environment, where the size of the gap is of the same order of magnitude as the body length of an ant microbot. In detail, the ant and the gap lengths are 97 μm and 120 μm, respectively, while the ant and the gap heights are both 20 μm (Fig. 4a). For a single robot performing forward locomotion and attempting to cross the gap, most failures occur after the middle pair of legs fall into the gap because the center of gravity (CG) of the microbot is located above the middle pair of legs. From this, we deduce that the maximum length of the gap that a single microbot can traverse is approximately equal to the distance $D_1$ from its CG to the front of the mandibles, which is 50 μm. Using the same principle, we derive that the maximum gap lengths that two and three 180° connected microbots can traverse are 93 μm and 145 μm, respectively (Fig. 4b). That is, a single ant microbot and two assembled microbots cannot pass through this gap, while three connected microbots are able to traverse successfully. The experimental results confirm our speculation, as shown in Fig. 4c–e (Supplementary Movie 9). When an individual microbot attempts to traverse the gap guided by the magnetic field, the robot falls completely into the gap because the length of the gap is larger than the body length of the robot (Fig. 4c). For the two connected microbots attempting to cross the gap, they fail because the distal edge of the gap is beyond the reachable space of the front legs thus causing the front legs to fall into the gap. The whole rotates on their latter unit legs and the former unit tilts forward, which results in a sloped body posture. Even so, they remain as one complete individual, indicating a high degree of assembly stability (Fig. 4d). By contrast, the three assembled microbots can pass through without difficulty. Three connected microbots will shift the CG far enough to the rear and provide a proper weight distribution to traverse the gap and the pitch angle of the body is always maintained close to zero during the crossing process. This demonstrates that the assembled microbots have a stronger ability to traverse the gap compared to an individual microbot (Fig. 4e).

## Maze navigation and drug delivery by microbot collectives with different assembled morphologies

Ant colonies in nature contain many individuals, which can collaborate to accomplish various tasks. In contrast, it is difficult to use our assembly strategy for large numbers of individuals. Our proposed method can accomplish the reconfigurable assembly of several microbots, and it is suitable for manipulating particles, cells, and other micro-objects whose sizes are concentrated in the range of a few micrometers to tens of micrometers. Benefiting from the proposed strategy of reversible assembly, we achieve cargo transportation and maze navigation by switching between I shape and C shape consisting of three magnetic units (Supplementary Fig. 21 and Supplementary Movie 10). The width of the entrance and exit of the maze allow only an I-shaped three-unit microbot to pass through and the entrance and exit of the maze have a gripping point to facilitate the switching of the assembly pattern. As illustrated in Supplementary Fig. 21, first, the three-unit microbot is manipulated into the maze in an I shape by a magnetic field, after which a light field is combined to enable it to clamp the gripping point. Here, due to the global actuation of the magnetic field in a small area, the purpose of clamping is to selectively manipulate one unit without affecting neighboring units to adjust the assembly pattern of the three units. Next, again under the coordination control of the magnetic and optical fields, the three assembled units are transformed from I shape to L shape and then to C shape to be able to carry a cargo. Then, with the cargo in their grasp, the C-shaped three-unit microbot is transported magnetically to the exit of the maze. Afterwards, the C-shaped three-unit microbot is reconfigured to an L shape and then to the initial I shape under the coordinated control of the magnetic and optical fields to release the cargo as well as to be able to exit the maze. Finally, the three-unit microbot assembled in the I shape is successfully disengaged from the gripping point and maneuvers out of the maze under the manipulation of the magnetic and light fields. These results indicate that the environmental adaptability

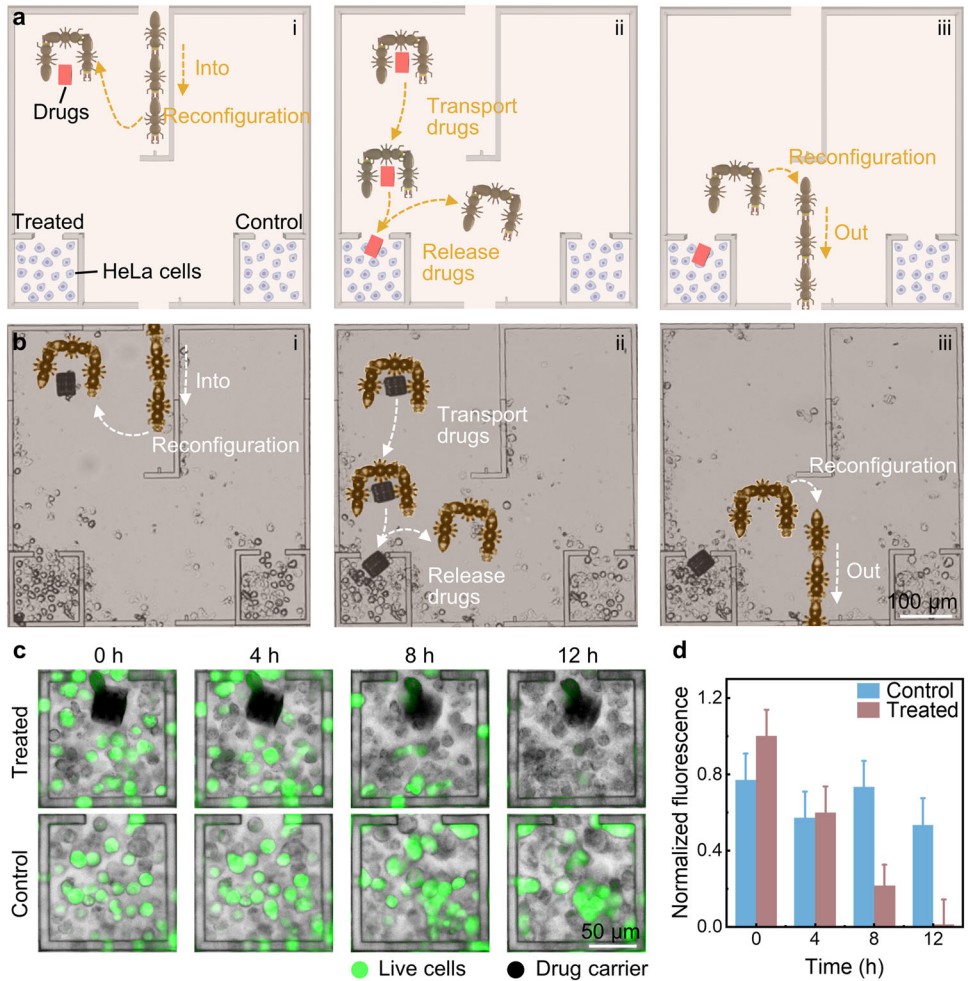

**Fig. 5 | Maze navigation and drug delivery by microbot collectives with different assembled morphologies. a** Illustration of experimental steps for accurate delivery of doxorubicin hydrogel block. The connected three microbots are initially located outside the entrance to the maze and are assembled in a 180° method so that they can be pulled through a narrow channel at the entrance using a magnetic field. Once inside the maze, they are reconfigured into a C-shaped morphology by switching from a 180° assembly method to a 90° assembly method under the coordinated control of the magnetic and light fields, allowing them to carry a carrier (red). The carrier is prepared by immersing a cubic hydrogel block processed by TPP in DOX solution (5 mg mL$^{-1}$) for 1 h to load drugs onto the carrier.

After the carrier is gripped, it is transported to the targeted position using a magnetic field. Afterwards, they are reconfigured into the I-shaped morphology in order to release the carrier. Finally, they pass through the narrow passage at the exit of the maze. **b** Optical images show the delivery process of doxorubicin hydrogel block. The images from each location have been stitched together to demonstrate the full path. **c** Fluorescence comparison images show the fluorescence changes of HeLa cells cultured in square micro-fences with and without doxorubicin hydrogel block inside in a time span of 12 h. **d** Quantitative statistics of fluorescence intensity of HeLa cells with and without doxorubicin hydrogel block inside. The error bars represent the standard error of the measurements ($n = 3$).

and functionality of microbots could be significantly improved by morphological reconfiguration, which is expected to facilitate the development of biomedical engineering.

Next, we use the drug carrier as delivery object to further investigate the transportation capabilities of the ant microbot collectives in a maze environment. Thanks to the proposed assembly strategy confers environmental adaptability and portability to the connected units, our microbots can perform drug delivery to Hela cells in the targeted region. As depicted in Fig. 5a, a photoresist square maze (600 μm in side length) with a square micro-fence (150 μm in side length) at each end of one edge is prepared by TPP process. The nutrient medium containing Hela cells is dropped onto the maze and cells outside the micro-fences in the maze are removed as much as possible. One of the micro-fences containing Hela cells is treated as the experimental group, i.e., the area where the drugs will be delivered to and function, while the other micro-fence serves as the reference group. In this process, the shape reconfiguration of microbot collectives and the transportation of the drug carrier are the same as for the cargo described in previous section. we manipulate the three-unit microbot to transport the cubic

drug carrier (20 μm in length) from the entrance of the maze to the experimental group by coordinating the magnetic and light fields, where the drug carrier is prepared by immersing a cubic hydrogel block processed by TPP in doxorubicin hydrochloride (DOX) solution (5 mg mL$^{-1}$) for 1 h to load drugs onto the carrier. Figure 5b shows time-lapse images of drug carrier delivery, and the images from each location have been stitched together to demonstrate the full path. After the carrier is delivered to the targeted position, the extent and intensity of the red fluorescence gradually increases over time and needs ~20 min to fill the entire area of the experimental group, indicating that the drugs had diffused rapidly from the carrier (Supplementary Fig. 22). During the 12 h of culture in the incubator, the green fluorescence of the cells in the experimental group gradually fades, while the cells in the control group maintain a strong green fluorescence all the time (Fig. 5c). Quantitative analysis reveals 98% attenuation of cells fluorescence intensity in the micro-fence containing the doxorubicin hydrogel block, whereas only 18% fluorescence attenuation is observed in the control group (Fig. 5d), demonstrating a significant inactivating effect of the drugs on the Hela cells.

## Discussion

Microbot collectives have a superior ability to accomplish complex tasks compared to individual microbots. Although many efforts have been invested in developing them[48–51], the lack of reliable connectivity between the micro units, the need of continuous input of external energy and the inability to independently control the designated unit have restricted their further advancement. The ant microbot collectives based on mechanically deformed assemblies designed in this study demonstrate assembly morphological diversity and strong stability and the ability to traverse gaps and environmental adaptability and portability under different assembly morphologies, which broaden the behaviors and functions of microbot collectives.

In terms of scenarios that specifically combine magnetic and optical control, we believe that fluid control in microfluidic chips is a suitable choice. For example, for microfluidic chips with multi-stage width channels, we can deploy I-shape microbot collectives in the main channel to act as dynamic microvalves. Meanwhile, for narrow channels with smaller widths, we can disassemble the ant microbot collectives and deploy only a single microbot in the narrow channel. In our proposed assembly strategy, a gripping point is required to assist in the transformation of the assembly shape. When in vitro, shape reconfiguration can be achieved with the help of square, round, triangular or even irregularly shaped gripping points in the environment.

In summary, inspired by the behavior of ant colonies, we develop a magnetically responsive ant microbot whose mandibles can be opened under the control of a light field. Combined with the design of its tail, the ant microbot collectives are capable of accurate assembly, dispersion and reconfiguration by mandibles clamping the tail under the synergistic control of magnetic and light fields, and the assembled morphologies based on mechanical deformation have remarkable robustness and are maintained in the absence of a continuous supply of external stimuli. In addition, the strategy achieves not only the assembly of magnetic units, but also the assembly between nonmagnetic units with the help of a magnetic unit. Afterwards, we present the ability of the assembled ant microbots to traverse a one-body-length gap. As a proof-of-concept application, three unit-assembled microbots are capable of switching assembly forms to adapt to their environment and perform transportation of microcargo and medication is demonstrated. There is no doubt that microbot collectives have multimodal and multifunctional capabilities compared to individual microbot. However, our strategy can currently only allow for assembly in a two-dimensional plane. Therefore, the introduction of other physical fields to accomplish the assembly between micro units may be a better choice. In the future, we believe that the microbot collectives will become powerful tools in the field of particle manipulation and drug transportation.

## Methods

### Materials

The Iron (II) chloride tetrahydrate ($FeCl_2 \cdot 4H_2O$), Iron (III) chloride hexahydrate ($FeCl_3 \cdot 6H_2O$), Ammonia water ($NH_3 \cdot H_2O$, 25 wt.%), Oleic acid and n-Octane were purchased from Sinopharm Chemical Reagent Co., Ltd. N-isopropylacrylamide, N, N'-methylenebis (acrylamide), Diphenyl (2,4,6-trimethylbenzoyl) phosphine oxide, ethylene glycol, Polyvinylpyrrolidone K30, Rhodamine 6 G, trisodium citrate, aqueous ammonia, and silver nitrate were purchased from Aladin.

### Preparation of magnetic photoresist

$FeCl_3 \cdot 6H_2O$ (2.69 g, 0.1 mol) and $FeCl_2 \cdot 4H_2O$ (0.99 g, 0.05 mol) were dissolved in DI water (100 mL) and then stirred for 30 min. Next, $NH_3 \cdot H_2O$ (25 wt.% 4 mL) was added dropwise to the mixed solution under vigorous stirring, resulting in the formation of $Fe_3O_4$ nanocrystals in weak alkaline solution. After stirring for 3 h, the resulting precipitate was separated from the mixture using a strong magnet and washed three times with DI water. The surfactant Oleic acid (0.61 g, 2.15 mmol) was added dropwise to the $Fe_3O_4$/water suspension at 80 °C under nitrogen protection with continuous stirring for 30 min. The prepared surface-modified $Fe_3O_4$ nanocrystals were then uniformly dispersed in n-Octane (3.2 mL, 20 mmol). Finally, after commercially available photoresist SZ2080 was added, the mixture was ultrasonic stirred for 1 h at room temperature to obtain magnetic photoresist.

### Preparation of silver precursor

Firstly, 14.1 mg of silver nitrate aqueous and 13.4 mg of trisodium citrate were added to 1 mL of DI water and stirred to mix the mixture well. Subsequently, $NH_3 \cdot H_2O$ (25 wt.%) was added dropwise to the mixture under stirring until a clarified solution was formed.

### Preparation of pNIPAM precursor

The precursor was prepared by dissolving 400 mg of N-isopropylacrylamide, 30 mg of N, N'-methylenebis (acrylamide), 30 mg of Diphenyl (2,4,6-trimethylbenzoyl) phosphine oxide, and 50 mg of Polyvinylpyrrolidone K30 into 450 μL of ethylene glycol, followed by the addition of 2 mg of Rhodamine 6 G for fluorescence imaging.

### Fabrication of 3D microstructures

A mode-locked Ti: sapphire ultrafast oscillator (Chameleon Vision-S, Coherent) with an oil immersion objective (60 ×, NA = 1.35) was used for both photopolymerization and photoreduction. The magnetic photoresist was first added dropwise to a coverslip and heated at 100 °C for 1 h. After that, the coverslip was placed on a high-precision platform, where the scanning oscillator in the XY plane and the Z-directional nano-position stage can be controlled to achieve point-by-point scanning processing in the magnetic photoresist. The processing files were obtained by converting the 3D models (SolidWorks) imported into the home-built software, which was able to control coordinately the position of the XY plane scanning oscillator and Z-directional piezoelectric stage. Subsequently, the sample was immersed in the developing solution (1-propanol) for 15 min to remove the uncured photoresist. In the second step, the hydrogel precursor was drop-added onto the same coverslip where there were previously processed microstructures and the bilayer hydrogel joints was fabricated to connected the ant mandibles and body. The complete ant microbot was obtained after 15 min of development with ethanol. In the final step, the silver precursor was dropped on the microstructures and the Ag layer can be deposited on any desired surface of the micro-ant by the photoreduction process.

### Cell culture

HeLa-EGFP cells were obtained from American Type Culture Collection and cultured in a humidified atmosphere at 37 °C with 5% $CO_2$. The cells were cultured in normal Dulbecco's modified eagle medium (Gibco, Thermo Fisher Scientific, Grand Island, NY) supplemented with 10% fetal bovine serum (HyClone, Logan, UT) and 1% penicillin/streptomycin (Gibco, Life Technologies, Grand Island, NY). For cells therapeutic experiment, Hela cells were detached from the culture dish by trypsinization with 0.25% Trypsin-EDTA (Gibco, Thermo Fisher Scientific, Grand Island, NY) at 37 °C for 30 s and centrifuged at 150 g for 5 min. Then, the cells were resuspended in culture medium at a density of $2 \times 10^6$ mL$^{-1}$ for use.

### Characterization of samples

The SEM images were taken by using a secondary-electron SEM (EVO18, ZEISS), and the samples were subjected to supercritical drying and coated with a gold film (a thickness of ~3−5 nm) beforehand. The optical microscopy images were obtained using an inverted microscope (DMI 3000B, Lecia). The energy dispersive spectrometer (EDS) analysis spectroscopy images were obtained by a COLD FESEM (Hitachi, SU8220).

## Simulation

The numerical simulation was performed in Comsol Multiphysics 5.4. The process of ant microbot mandibles opening in response to the light field involved heat transfer, mass diffusion and large deformation, which was proposed by He et al.[52] Details of the simulation methods were reported in Supplementary Information.

## Data availability

All data needed to evaluate the conclusions in the paper are present in the manuscript and Supplementary Information. The data are also available upon request from the corresponding author. Source data are provided with this paper.

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

## Acknowledgements

This work was supported by the National Natural Science Foundation of China (Nos. 61927814, 62325507, 52122511), the National Key Research and Development Program of China (No. 2021YFF0502700), Major Scientific and Technological Projects in Anhui Province (202203a05020014). We acknowledge the Experimental Center of Engineering and Material Sciences at USTC for the fabrication and measuring of samples. This work was partly carried out at the USTC Center for Micro and Nanoscale Research and Fabrication.

## Author contributions

Z.R. and C.X. conceived the idea and designed the project. Z.R., K.L., H.W., D.W. (Dawei Wang), and L.X. performed the experiments and the characterization. Z.R. completed data analysis and figure depicture. Z.R. wrote and revised the paper. Y.H., J.L., J.C., and D.W. (Dong Wu) supervised the project.

## Competing interests

The authors declare no competing interests.
