## [Peer Review File · Nature Communications]

REVIEWER COMMENTS

Reviewer #1 (Remarks to the Author):

The manuscript titled "Ant-inspired reconfigurable microbot collectives" discusses an innovative technique for enabling reconfigurable microbots through a combination of magnetic and light-induced actuation. The magnetic component is utilized for movement and orientation, while the light-induced mechanism facilitates the operation of the microclaw, enabling inter-robotic connection.

The visual aids, including figures and videos, effectively elucidate the concept, greatly enhancing the manuscript's comprehensibility. The fabrication technique, while intriguing, appears to have been previously published by the same group.

To further enhance the manuscript, I recommend the authors consider the following suggestions:

Application Scenarios: It would be beneficial to delineate specific scenarios where this mixed actuation technique could be advantageous. While laser light offers high directionality, resolution, and energy, enabling substantial control degrees of freedom, its inability to penetrate objects contrasts with the advantages of magnetic control. Clarifying scenarios where a combination of magnetic and light control is particularly effective would strengthen the manuscript's applicability.

Agent Differentiation: Differentiating magnetic and non-magnetic agents through color coding would aid in reader comprehension. The manuscript employs a mix of non-magnetic agents and fixed environmental gripping points, as observed in the "maze navigation and cargo transport" video. However, this raises questions about the practicality of such an approach in real-world applications, particularly in medical scenarios as depicted in Figure 5. Addressing the limitations and logistical considerations, such as the manual distribution of non-magnetic agents, would provide valuable clarity.

Design Considerations: While the concept draws inspiration from ants, the current robot design, particularly the leg structures, does not seem to offer functional advantages. Reconsidering the design to emphasize functionality over form could enhance the manuscript's focus on practical applications.

Video Demonstration: Given that the results in Figure 5 represent a culmination of the research, including a comprehensive video demonstration would significantly augment the impact of these findings.

I look forward to re-reviewing the manuscript upon the incorporation of these suggestions.

Reviewer #2 (Remarks to the Author):

This paper presents 97µm robot ants guided using magnetic fields, and actuated using light to enable the opening of claws used to connect micro robot ants together. Using this mechanism, the ants can assemble, disassemble and reconfigure.

The combination of magnetic fields and light to control collectives, as well as the robot design are novel and will be of interest to the community.

Results show the robots assembling into different shapes, enabling functions such as traversing a gap, or carrying a drug payload to a cancer cell.

Overall the paper is well written, and nicely situated in the state-of-the-art. The methods are clear. The videos are exciting and clearly show the results.

The following points should be considered to improve the manuscript.

Although the ant shape is fun to watch, it's not clear why it makes sense to use an ant shape for this work. Ants do not operate at such small scales (size), and they work in much larger numbers. Would the design be more streamlined if you had a rod with a claw, or a disc with a hook? The ant shape is slightly gratuitous.

One large drawback of the system is that it relies on sequential assembly of the structures, and often tethering of one of the ants. This is due to the need to locomote individual ants, and then actuate their claw in the right sequence. This makes it challenging to scale the solutions to large numbers of ants - limiting the collective nature of the work. Additional discussions on this point would be helpful.

The cancer scenario is not credible and the claims that this approach can be used to deliver therapeutic cargos in the context of cancer are exaggerated. For this approach to work you would need large numbers of cargos, and the robots would need to be able to navigate the body, and in complex tumour environments. Such navigation would be very different from the demonstrated maze environment with perfect visibility. It would be best to tone down claims about cancer treatment, and focus on cargo transport. Or perhaps more context could be given so the reader can easily understand how this would be used in the context of cancer treatment.

Figure 1c looks fake - the ants are too big compared to the apple.

Reviewer #3 (Remarks to the Author):

Review of "Ant-inspired reconfigurable microbot collectives"

The key contribution is a light-activated, default-closed gripper on a micro block. The block is equipped with a single attachment block, shaped to allow assembly at 90 degree increments. The design only allows chains, and the demonstrations show chains of up to five blocks (Fig. 3C, the 'S').

Some blocks are printed with a magnetic photoresist, so they can be manoeuvred and oriented by the magnetic field. When this block is attached to one or more other blocks, the chain can be manoeuvred. The blocks are printed (Two photon process) in the form of a ant that is 0.1mm long. The ant shapes are visually appealing. These 'ants' are passive blocks, manipulated by an external magnetic field for propulsion and an external directed light source that can open the jaws.

This is a fun contribution. The ability to lock individual components in at 0/90/180/270 degree angles to make rigid attachments using a default closed gripper is clever. The experiments are interesting and the results are easy to understand.

"BIG ISSUES" In order of importance

How can this be extended or made parallel? Currently one robot can provide propulsive force, and the laser can open one jaw at a time. How does this scale? By this question:

(a) can multiple grippers be actuated in unison?

(b) can multiple robots be moved simultaneously?

(c) can movement speeds be differentiated by robot design so that some robots move faster than others under control of the same magnetic field? If so, then certain robots could be controlled to latch onto parts of the environment

Is the ant form optimized in any manner?

Why only a single attachment point and a single gripper? Ants can grab with mouth and their 6 feet, and can attach at many points along the legs and head. the current construction only allows chains. Adding multiple attachment points and grippers would enable tree structures and more rigid connections. Using smooth attachment points would enable pivot points.

"MINOR EDITS"

In title, "Ant-inspired" seems too much. The components are ant-shaped, but all the tasks shown could be done by a rectangular block with a gripper and an attachment point.

Fig. 1: please show a schematic of the ant, with dimensions. Where are the grasp points?

line 217: What does it mean that "assembled microstructures are destroyed."? Does this only mean that at least one ant is separated, or does deformation of the structure also count?

How is the manipulating magnetic field moved? Is the magnet held under the workspace or above it? Is the magnet moved by a stage or by hand?

Line 30--31: Shrews don't intertwine, they bite onto the base of the tail of another "a conga line".

The paper says 'claws' many times, but these should all be "mandibles". Claws are on feet.

Line 141, if the jaws only take 20ms to open and close, please run 1000 cycles (which should take around 20 s). This will give a better indication of the longevity of the part.

Line 153: "the ant microbot rotates sensitively and rapidly excellent controllability"  this is qualitative and not informative. Please instead give the standard measures of rise time, steady-state error, and overshoot.

It is unclear how the magnetic field is applied. The pictures show a single permanent magnet. Is this automated or manual? Is there a motor or xy stage to move this magnet?
Would it work just as well with a set of electromagnets?

Fig. 3, which ants are magnetic? (Are all of them?)

Fig. 3 e and f, "Frequence" -> "Frequency"

"TINY EDITS" (e.g. typos)

Figure 1C is a compiled ('fake') photo of three ants rolling a gooseberry fruit. Please replace with an actual photo of real ants.

<https://depositphotos.com/photo/ants-carrying-food-together-teamwork-concept-162201840.html>

Line 13 "individual. However, current various weak force induced microbot collectives still face the" is too many adjectives. Link words modified with a dash: "force-induced"? What is a 'weak force'?

Line 15: "imprecisely individual control" -> "imprecise individual control"

Line 21 "to cross a large gap" is imprecise. Better is "to cross a one-body-length gap"

Line 22: "transport a microcargo." -> "transport microcargo."

Line 22 "the field of clustered microbots functions," no idea what this is. maybe "the abilities of microbots,"?

Line 28 "One type, in which there is no direct contact between individuals" -> "In the first category there is no direct contact between individuals"

Line 30 "contrast, the other, more stable collectives" -> "contrast, more stable collectives"

Line 31 "These collectives formed" -> "These collectives are formed"

Line 32 "locking into each" -> "locking onto each"

Line 38-39 "individually actuated for specified unit" -> "individually actuated".

Line 46 "controlled and steady manner." -> "controlled and stable manner."

Line 50 "predefined director field orientations" what is this? "predefined field orientation directions" ?

Line 110 "and instantaneous contraction can be observed"  8 ms is not instantaneous. -> "and full contraction in 8 ms is observed"

pg 6, what are the units for SRT?

line 125: "temperature of joint is" -> "temperature of the joint is"

line 129 "actuated by the joints to become the opening state." -> "actuated by the joints to open."

line 157 "As the schematics shown in Figure 3a," -> "As the schematics show in Figure 3a,"
line 184 "number of the two assemble modes," -> "number of the two assembly modes,"
line 192 "Besides that, the" -> "The"
Line 207 "configurations that close resemblance to the tetrominoes" -> "configurations that closely resemble the tetrominoes"
Line 226 : add the ant height and length so the reader can see the correlation.
Line 229: "the center of gravity (CG) of the microbot is located in the middle pair of legs" -> "the center of gravity (CG) of the microbot is located above the middle pair of legs"
Line 252 "It is worth noting that the width" -> "The width"
Lines 281 and 283 "drugs carrier" -> "drug carrier"
Line 285 "gradually increase over" -> "gradually increases over"
Fig. 5, for (n=3), please make the n italic.

VIDEO:

Supplementary material: the same icon for a laser is used in Fig. 1 a and b. Is the same wavelength used? (also Fig. 2a)

Fig. 10 is hard to see the difference in colorings. Are the orientations of the magnetic material the same in each ant? If so, in Fig. 10 C , column2, how are the three magnetic photoresist ants not all in the same orientation?

Fig. 11, please add time stamps to the images.

Article File PDF (1393KB) Source File (DOCX) 4207KB

Supplementary Information PDF (1570KB) Source File (DOCX) 4769KB

Video - Supplementary Movie 1 Video (5474KB) Source File (MP4) 5474KB -- tiny 'ant' that opens and closes jaws. Scale bar at bottom right

Video - Supplementary Movie 2 Video (4142KB) Source File (MP4) 4142KB -- tiny 'ant' (same as video 1?) that opens and closes jaws 50x. Scale bar at bottom right

Video - Supplementary Movie 3 Video (4122KB) Source File (MP4) 4122KB -- one ant approaches another and grasps it. Unclear why the first ant does not move. Also hard to tell when light is on -- better to always have an annotation that says light is off. Also hard to tell which movement is due to the camera moving and which is due to tiny robot moving. Adding a stationary landmark would help.

Video - Supplementary Movie 4 Video (6125KB) Source File (MP4) 6125KB -- ant approaches at 90 degrees. What is the ant grasping? Is the B field the orientation of the field, or the orientation of the gradient, or both?

Video - Supplementary Movie 5 Video (5343KB) Source File (MP4) 5343KB -- two-chain ant approaches 3rd ant from back. What is the radius of the light beam? Where is the focus? (It can make a solitary ant open or close its jaws.)

Video - Supplementary Movie 6 Video (5126KB) Source File (MP4) 5126KB --two-chain ant approaches 3rd ant from 90-deg.

Video - Supplementary Movie 7 Video (4566KB) Source File (MP4) 4566KB -- two-chain ant approaches and connects to 3rd ant from back, then back ant separates and moves away.

Video - Supplementary Movie 8 Video (14223KB) Source File (MP4) 14223KB -- 'u'-shaped 3 ant formation is placed on a shake table? (how does that work?) Misspelling 'Untab'e" -> "Unstable" at 360 micrometer amplitude.

Video - Supplementary Movie 9 Video (7911KB) Source File (MP4) 7911KB -- gap traversal. Please show the magnetic field. What is the height of the gap? A schematic showing a side view would help immensely. (show different illustration for all 3 tests, and show the failure case.

Video - Supplementary Movie 10 Video (19197KB) Source File (MP4) 19197KB -- three ants reconfigure into a 'u' shape and move a square object.

This last video is composed of many takes, and the square box jumps around, often from a bad position to a 'good' position in the next take. How many trials were conducted and spliced together to make this video? The video at present is unacceptable and misleading. At minimum, the trial number for each video segment needs to be listed.

A schematic showing a side view would help. Which robot(s) are magnetic?

Reply to the Reviewers' and editor's comments

Thanks a lot for the comments. The manuscript (Manuscript ID: NCOMMS-23-49483) has been carefully modified according to the comments. The revisions were marked in red as shown below. We appreciate the valuable comments and suggestions from the reviewers and editor. The point-to-point answers to the comments of the reviewers and editor are listed as follows. There is also a change list in the end. (The numbers of Page and line in "Revision list" correspond to "Manuscript with detailed changes.pdf" and "SI with detailed changes.pdf")

Reviewer #1 (Remarks to the Author):

The manuscript titled “Ant-inspired reconfigurable microbot collectives” discusses an innovative technique for enabling reconfigurable microbots through a combination of magnetic and light-induced actuation. The magnetic component is utilized for movement and orientation, while the light-induced mechanism facilitates the operation of the microclaw, enabling inter-robotic connection.

The visual aids, including figures and videos, effectively elucidate the concept, greatly enhancing the manuscript’s comprehensibility. The fabrication technique, while intriguing, appears to have been previously published by the same group.

To further enhance the manuscript, I recommend the authors consider the following suggestions:

Response: Many thanks to the reviewer for careful review and evaluation of this work. Your constructive comments have helped us to further improve our manuscript.

1. Application Scenarios: It would be beneficial to delineate specific scenarios where this mixed actuation technique could be advantageous. While laser light offers high directionality, resolution, and energy, enabling substantial control degrees of freedom, its inability to penetrate objects contrasts with the advantages of magnetic control. Clarifying scenarios where a combination of magnetic and light control is particularly effective would strengthen the manuscript’s applicability.

Response: We sincerely thank you for your great advice. As we all know, magnetic and light controls each have their own unique advantages (*Adv. Mater.* **32**, 1906766 (2020)). On the one hand, magnetic control exhibits remote and rapid motion control, deep tissue penetration, and high biosafety. However, the precise manipulation of one component in a group of magnetic components is still very difficult for magnetic control. On the other hand, light control exhibits high resolution, high degrees of freedom, and single-component manipulability. However, the limited tissue penetration capability still restricts its application in biological organisms. In addition, although the light control

can realize the motion, the movement speed is too slow to adapt to practical applications. Therefore, combining magnetic and light control can compensate for each other's shortcomings. In this work, we use the magnetic field to control the motion and orientation of the microbot, and the light field to control the opening of the microbot's mandibles for subsequent assembly.

We understand the reviewer's concern about the weak penetration of the light field, which makes it difficult to be used in opaque environments. On the one hand, light-controlled microbots can be used in lab-on-a-chip applications (cell collection and sorting). On the other hand, optical fibers can be used to direct light into opaque media to control the microbots to complete the functionalities. In terms of scenarios that specifically combine magnetic and optical control, we believe that fluid control and cell manipulation in microfluidic chips is a suitable choice. For example, for microfluidic chips with multi-stage width channels, we can deploy I-shaped microbot collectives in the main channel to act as dynamic microvalves (Figure R1.1a). Meanwhile, for narrow channels with smaller widths, we can disassemble the ant microbot collectives and deploy only a single microbot in the narrow channel (Figure R1.1b). In addition, light and magnetic controls are advantageous for in vitro manipulation of cells of different sizes. For example, when performing in vitro fertilization, it is necessary to manipulate small-sized sperm ($50 \times 6 \mu\text{m}$) and large-sized oocyte ($200 \times 200 \mu\text{m}$). In this way, a single microbot can manipulate sperm, where the light field is used to control the mandibles to grab the sperm and the magnetic field is used to transport it (Figure R1.2a). Subsequently, the C-shaped microbot collectives constructed by light and magnetic fields can be used to manipulate larger-sized oocyte (Figure R1.2b). We believe that the proposed reconfigurable microbot collectives could perform the above tasks through the combination of magnetic and light control. Moreover, we will also discuss further clinical applications of these microbot collectives with biological and medical experts.

Figure R1.1 Schematic of I-shaped microbot collectives for microfluidic applications.

Figure R1.2 Schematic of C-shaped microbot collectives for in vitro manipulation of cells of different sizes.

Reference:

Sitti, M. & Wiersma, D. S. Pros and cons: Magnetic versus optical microrobots. *Adv. Mater.* **32**, 1906766 (2020).

List of Revisions:

✓ We added the corresponding description on page 16 line 328: “*In terms of scenarios that specifically combine magnetic and optical control, we believe that fluid control and cell manipulation in microfluidic chips is a suitable choice. For example, for microfluidic chips with multi-stage width channels, we can deploy I-shape microbot collectives in the main channel to act as dynamic microvalves. Meanwhile, for narrow channels with smaller widths, we can disassemble the ant microbot collectives and*

deploy only a single microbot in the narrow channel. In addition, light and magnetic controls are advantageous for in vitro manipulation of cells of different sizes. For example, when performing in vitro fertilization, it is necessary to manipulate small-sized sperm and large-sized oocyte. In this way, a single microbot can manipulate sperm, where the light field is used to control the mandibles to grab the sperm and the magnetic field is used to transport it. Subsequently, the C-shaped microbot collectives constructed by light and magnetic fields can be used to manipulate larger-sized oocyte.”

2. Agent Differentiation: Differentiating magnetic and non-magnetic agents through color coding would aid in reader comprehension. The manuscript employs a mix of non-magnetic agents and fixed environmental gripping points, as observed in the “maze navigation and cargo transport” video. However, this raises questions about the practicality of such an approach in real-world applications, particularly in medical scenarios as depicted in Figure 5. Addressing the limitations and logistical considerations, such as the manual distribution of non-magnetic agents, would provide valuable clarity.

Response: We thank the reviewers for valuable comments. As described by the reviewers, we use a mix of non-magnetic agents and fixed environmental gripping points to help the microbot collectives perform their functions. In this work, microbot collectives can be composed either entirely of magnetic components or of a mixture of non-magnetic and magnetic components. The different types of constituent components of the microbot collectives demonstrate the flexibility and wide material applicability of our proposed strategy. For the manual distribution of non-magnetic agents, we can deliver multiple non-magnetic components to different target locations individually by a magnetic microbot, where the process is shown in Figure R1.3, the magnetic microbots guide the orderly assembly of microbots composed of different non-magnetic materials.

In the conceptual demonstration of the medical application in Figure 5, all the ant microbots are magnetic. When the microbot collectives grasp an environmental

gripping point, we can sequentially manipulate the microbots dissembled from the collectives to reconfigure the I-shaped microbot collectives into a C-shaped one. Furthermore, in addition to the designed rectangular gripping point, the microbots can perform grasping functions on different shapes of gripping points, such as circle, triangle, and semicircle (Figure R1.4).

Indeed, there are no pre-designed artificial gripping points in living organisms. However, the surface of tissues in living organisms often have microfold cells of various shapes (*Mucosal Immunology*, **6**, 666-677 (2013)), such as the stomach, small intestine, and bile ducts, which can serve as natural gripping points for assisting the microbot collectives to perform shape reconstruction. In addition, thanks to the flexibility of femtosecond laser two-photon polymerization to fabricate three-dimensional structures, we can also design microbots with a needle-like shape, which can penetrate and anchor in tissues (*Advanced Healthcare Materials* **9**, 1901697 (2020)). In this way, the needle-like microbot can form an artificial gripping point in vivo for the microbot collectives to accomplish shape reconstructions and functions.

Figure R1.3 The time-sequence optical images of the process of assembling five individual units containing four different materials. Scale bar, 100 μm .

Figure R1.4 Optical images of different shaped gripping points.

References:

1. Mabbott, N. A., Donaldson, D. S., Ohno, H., Williams, I. R. & Mahajan, A. Microfold (M) cells: important immunosurveillance posts in the intestinal epithelium. *Mucosal Immunology*. **6**, 666-677 (2013).
2. Lee, S. et al. A needle - type microrobot for targeted drug delivery by affixing to a microtissue. *Advanced Healthcare Materials* **9**, 1901697 (2020).

List of Revisions:

✓ We added the corresponding description on page 16 line 339: *“In our proposed assembly strategy, gripping point is required to assist in the transformation of the assembly shape. When in vitro, shape reconfiguration can be achieved with the help of square, round, triangular or even irregularly shaped gripping points in the environment. Indeed, there are no pre-designed artificial gripping points in living organisms. However, the surface of tissues in living organisms often have microfold cells of various shapes, such as the stomach, small intestine, and bile ducts, which can serve as natural gripping points for assisting the microbot collectives to perform shape reconstruction. In addition, thanks to the flexibility of femtosecond laser two-photon polymerization to fabricate three-dimensional structures, we can also design microbots with a needle-like shape, which can penetrate and anchor in tissues. In this way, the needle-like microbot can form an artificial gripping point in vivo for the microbot collectives to accomplish shape reconfigurations and functions.”*

3. Design Considerations: While the concept draws inspiration from ants, the current robot design, particularly the leg structures, does not seem to offer functional advantages. Reconsidering the design to emphasize functionality over form could

enhance the manuscript's focus on practical applications.

Response: We sincerely thank the reviewer for your comments. The reviewer is right that the microbot collective strategy proposed in this work is equally applicable to structures other than ant-like microbots. In our work, the leg structures can greatly reduce the contact area between the microbot and the substrate, which in turn decreases the friction of the substrate on the microbot making the microbot move more smoothly. In addition, according to the suggestion of reviewer #3, we believe that the leg structures can also be used as gripping points for more complex assemblies. Due to the flexibility of our fabrication approach, we design different shape microbots and perform new experiments. As shown in Figure R1.5a, we first simplify the body of the microbot into a rectangular shape, which can also be reversibly reconfigured into C, O, V, S, and W shapes by synergistic control of the magnetic and light fields (Figure R1.5b). Then, we add claws and gripping points around the rectangular body. As shown in Figure R1.6a, a microbot has two claws and six legs (six gripping points, Figure R1.6b). In this way, microbot collectives can be constructed into more complex bifurcated tree-like patterns (Figure R1.6c). In addition, we have developed simple disk-shaped microbots with two claws and two gripping points (Figure R1.7a), which can be assembled into shapes dispersing from the center outward (Figure R1.7b). We believe that the above shape optimization and complex pattern reconstruction of microbot collectives can meet the needs of more complicated applications.

Although our fabrication and reconfiguration strategies can be applied to microbots of different shapes, our work is inspired by the behavior of ant colonies in nature. Thus, we choose ant-shaped microbots as the typical structure for our research work. Moreover, the ant-shaped microbot collectives' behaviors of crossing the gap and transporting heavy micro-objects are also verified, which perfectly matches the behavior of real ant colonies. Therefore, to maintain the source of inspiration and integrity of the study, we want to keep the ant-shaped microbots in this work. We hope and appreciate that the reviewers can understand the preferences and reasons for choosing ant-shaped microbots.

Figure R1.5 Simplified rectangular-shaped microbot. (a) Schematic diagram and optical image of a rectangular-shaped microbot. (b) Optical images of C, J, O, V, S, and W shape assembled from rectangular-shaped microbots.

Figure R1.6 A microbot with two claws and six legs. (a) Schematic diagram and optical image of the microbot. (b) Optical images of each leg of the microbot used as a gripping point for assembly. (c) Optical diagrams of complex bifurcated tree-like patterns assembled from the microbots.

Figure R1.7 Disk-shaped microbot. (a) Schematic diagram and optical image of a disk-shaped microbot. (b) Optical diagrams of shapes dispersed from the center outward assembled from disk-shaped microbots.

List of Revisions:

✓ We added the corresponding description on page 10 line 197: *“In addition to the ant microbot, our assembly strategy is equally applicable to other shaped microbots with multiple claws and gripping points. Thanks to the flexibility of our fabrication approach, we have also designed other shaped microbots and realized more complex assembly forms (Supplementary Figure 13). For example, rectangular-shaped microbots with two claws and six gripping points can be constructed into bifurcated tree-like patterns (Supplementary Figure 14), and disk-shaped microbots with two claws and two gripping points can be assembled into shapes dispersing from the center outward (Supplementary Figure 15).”*

✓ We added Supplementary Figure 13-15 and the corresponding description in the supplementary materials.

4. Video Demonstration: Given that the results in Figure 5 represent a culmination of the research, including a comprehensive video demonstration would significantly augment the impact of these findings.

Response: We thank you very much for the reviewer’s suggestion. Combining the comments of reviewer #2 and reviewer #3, the microbot collectives for drug delivery and cancer cell therapy is more suitable as a proof of concept for an application rather than specifically emphasizing it. This work would like to highlight the microbot

collectives for micro-maze traversal and heavy object transportation through shape reconfiguration. The whole process has been represented in Supplementary Video 10. Considering that the microbot collective shape reconfiguration strategy, maze terrain traversal, and object transportation are fully consistent with the cell therapy application section, we believe that Supplementary Video 10 is sufficient to help readers understand the entire cancer cell therapy process.

List of Revisions:

- ✓ We added the corresponding description on page 15 line 303: *“In this process, the shape reconfiguration of microbot collectives and the transportation of the drug carrier are the same as for the cargo described in previous section.”*

Reviewer #2 (Remarks to the Author):

This paper presents 97 μm robot ants guided using magnetic fields, and actuated using light to enable the opening of claws used to connect micro robot ants together. Using this mechanism, the ants can assemble, disassemble and reconfigure.

The combination of magnetic fields and light to control collectives, as well as the robot design are novel and will be of interest to the community.

Results show the robots assembling into different shapes, enabling functions such as traversing a gap, or carrying a drug payload to a cancer cell.

Overall the paper is well written, and nicely situated in the state-of-the-art. The methods are clear. The videos are exciting and clearly show the results.

The following points should be considered to improve the manuscript.

Response: Thank you very much for your positive comments. We have answered each comment and made the appropriate changes in the revised version. Please review the details of the point-by-point responses below.

1. Although the ant shape is fun to watch, it's not clear why it makes sense to use an ant shape for this work. Ants do not operate at such small scales (size), and they work in much larger numbers. Would the design be more streamlined if you had a rod with a claw, or a disc with a hook? The ant shape is slightly gratuitous.

Response: We sincerely thank the reviewer for your helpful suggestion. The reviewer is right that the microbot collective reconfiguration strategy proposed in this paper is equally applicable to structures other than ant-like microbots. According to the reviewers' comments, we perform some new experiments to verify the structure design flexibility of our approach. As shown in Figure R2.1a, we first simplify the body of the microbot into a rectangular shape, which can also be reversibly reconfigured into C, O, V, S, and W shapes by synergistic control of the magnetic and light fields (Figure R2.1b). Then, we add claws and gripping points around the rectangular body. As shown in

Figure R2.2a, a microbot has two claws and six legs on its body, and each leg can be used as a gripping point (Figure R2.2b). In this way, microbot collectives can be constructed into more complex bifurcated tree-like patterns (Figure R2.2c). In addition, we have developed simple disk-shaped microbots with two claws and two gripping points (Figure R2.3a), which can be assembled into shapes dispersing from the center outward (Figure R2.3b). We believe that the above shape optimization and complex pattern reconstruction of microbot collectives can meet the needs of more complicated applications.

Although our fabrication and reconfiguration strategies can be applied to microbots of different shapes, this work is inspired by the behavior of ant colonies in nature. Thus, we choose ant-shaped microbots as the typical structure for our research work. What's more, the ant-shaped microbot collective's behaviors of crossing the gap and transporting heavy micro-objects are also verified, which perfectly matches the behavior of real ant colonies. Moreover, to maintain the source of inspiration and integrity of the study, we want to keep the ant-shaped microbots in this work. We hope and appreciate that the reviewers can understand the preferences and reasons for choosing ant-shaped microbots.

Figure R2.1 Simplified rectangular-shaped microbot. (a) Schematic diagram and optical image of a rectangular-shaped microbot. (b) Optical images of C, J, O, V, S, and W shape assembled from rectangular-shaped microbots.

Figure R2.2 A microbot with two claws and six legs. (a) Schematic diagram and optical image of the microbot. (b) Optical images of each leg of the microbot used as a gripping point for assembly. (c) Optical diagrams of complex bifurcated tree-like patterns assembled from the microbots.

Figure R2.3 Disk-shaped microbot. (a) Schematic diagram and optical image of a disk-shaped microbot. (b) Optical diagrams of shapes dispersed from the center outward assembled from disk-shaped microbots.

List of Revisions:

- ✓ We added the corresponding description on page 10 line 197: “In addition to the

ant microbot, our assembly strategy is equally applicable to other shaped microbots with multiple claws and gripping points. Thanks to the flexibility of our fabrication approach, we have also designed other shaped microbots and realized more complex assembly forms (Supplementary Figure 13). For example, rectangular-shaped microbots with two claws and six gripping points can be constructed into bifurcated tree-like patterns (Supplementary Figure 14), and disk-shaped microbots with two claws and two gripping points can be assembled into shapes dispersing from the center outward (Supplementary Figure 15)."

✓ We added Supplementary Figure 13-15 and the corresponding description in the supplementary materials.

2. One large drawback of the system is that it relies on sequential assembly of the structures, and often tethering of one of the ants. This is due to the need to locomote individual ants, and then actuate their claw in the right sequence. This makes it challenging to scale the solutions to large numbers of ants - limiting the collective nature of the work. Additional discussions on this point would help.

Response: We sincerely appreciate the reviewer for the insightful comments. The ant colonies or bee swarms in nature usually contain many individuals, which can collaborate to accomplish various tasks. We agree with the reviewers' statements that the strategy proposed in this work is difficult to realize large numbers (>20) of microbot assemblies. Our proposed method can accomplish the reconfigurable assembly of several microbots (Figure R2.2), and it is suitable for manipulating particles, cells, and other micro-objects whose sizes are concentrated in the range of a few micrometers to tens of micrometers. For larger objects (>500 μm), existing commercial microsystems are easy to manipulate them (*Annual Review of Biomedical Engineering* **11**, 203-233 (2009)). Therefore, the assembly of a large number of microbots is not a priority for our work. According to the reviewer's suggestions, we added some discussions to the revised manuscript.

Reference:

Kim, D.-H., Wong, P. K., Park, J., Levchenko, A. & Sun, Y. Microengineered platforms for cell mechanobiology. *Annual Review of Biomedical Engineering* **11**, 203-233 (2009).

List of Revisions:

✓ We added the corresponding description on page 13 line 269: *“The ant colonies in nature contain many individuals, which can collaborate to accomplish various tasks. In contrast, the number of individuals in the collectives formed by our proposed assembly strategy is difficult to be as large as that of ant colonies. Our proposed method can accomplish the reconfigurable assembly of several microbots, and it is suitable for manipulating particles, cells, and other micro-objects whose sizes are concentrated in the range of a few micrometers to tens of micrometers.”*

3. The cancer scenario is not credible and the claims that this approach can be used to deliver therapeutic cargos in the context of cancer are exaggerated. For this approach to work you would need large numbers of cargos, and the robots would need to be able to navigate the body, and in complex tumour environments. Such navigation would be very different from the demonstrated maze environment with perfect visibility. It would be best to tone down claims about cancer treatment, and focus on cargo transport. Or perhaps more context could be given so the reader can easily understand how this would be used in the context of cancer treatment.

Response: We thank the reviewer for this helpful suggestion. The reviewer is correct that the approach we propose in this work is particularly difficult to use for cancer treatment. It is currently more suitable for cargo transportation, or in vitro cell manipulation. Following the reviewers’ comments, we have removed cancer treatment-related content in the revised manuscript to tone down claims about cancer treatment.

List of Revisions:

✓ We revised the corresponding description on page 14 line 294: *“Next, we use the drug carrier as delivery object to further investigate the transportation capabilities of the ant microbot collectives in a maze environment.”*

- ✓ We changed the “*cancer cell*” to “*Hela cell*” on page 14 line 297 and page 15 line 318.
- ✓ We deleted the corresponding description on page 15 line 313: “*The treatment effect of DOX could be evaluated by observing the change of cell fluorescence in the two micro-fences.*”
- ✓ We deleted the corresponding description on page 15 line 318: “*which further demonstrates the targeted treatment of Hela cells.*”

4. Figure 1c looks fake - the ants are too big compared to the apple.

Response: We thank the reviewer for pointing this out. The photo in Figure 1c has been replaced by an actual photo of real ants (Figure R2.4).

Figure R2.4 Actual photo of real ants working together to carry.

List of Revisions:

- ✓ We revised Figure 1c.

Reviewer #3 (Remarks to the Author):

Review of “Ant-inspired reconfigurable microbot collectives”

The key contribution is a light-activated, default-closed gripper on a micro block. The block is equipped with a single attachment block, shaped to allow assembly at 90 degree increments. The design only allows chains, and the demonstrations show chains of up to five blocks (Fig. 3C, the “S”).

Some blocks are printed with a magnetic photoresist, so they can be manoeuvred and oriented by the magnetic field. When this block is attached to one or more other blocks, the chain can be manoeuvred.

The blocks are printed (Two photon process) in the form of an ant that is 0.1mm long. The ant shapes are visually appealing. These “ants” are passive blocks, manipulated by an external magnetic field for propulsion and external directed light source that can open the jaws.

This is a fun contribution. The ability to lock individual components in at 0/90/180/270 degree angles to make rigid attachments using a default closed gripper is clever. The experiments are interesting and the results are easy to understand.

Response: Many thanks to the reviewer for careful review and positive comments of this work. We have answered each comment and made the appropriate changes in the revised version. Please review the details of the point-by-point responses below.

“BIG ISSUES” In order of importance

1. How can this be extended or made parallel? Currently one robot can provide propulsive force, and the laser can open one jaw at a time. How does this scale? By this question:

- (a) can multiple grippers be actuated in unison?
- (b) can multiple robots be moved simultaneously?
- (c) can movement speeds be differentiated by robot design so that some robots move

faster than others under control of the same magnetic field? If so, then certain robots could be controlled to latch onto parts of the environment.

Response: We sincerely appreciate the reviewer for the constructive comment.

(a) In our work, the opening of the ant microbot mandible is driven by a light field. In general, a focused light spot can only effectively actuate one mandible opening. To actuate multiple mandibles to open, we can introduce a spatial light modulator (SLM) to generate multiple beams, which is suitable to drive multiple mandibles to open simultaneously. As shown in Figure R3.1a, we first verified the feasibility of this scheme using several ant microbots. When the hologram corresponding to only one focal point is loaded onto the SLM, the mandibles of only one ant microbot out of many ant microbots are stimulated to open by a single beam of light. Switching to a hologram corresponding to two foci, the mandibles of two ant microbots out of several ant microbots are stimulated to open by two beams of light at the same time. Similarly, when the SLM is loaded with a hologram corresponding to four foci, the mandibles of all ant microbots are open (Figure R3.1b). In addition, we design a multi-gripper microbot (Figure R3.1c) with four grippers around the body. Using the same actuation strategy, we realize that one of the four grippers of this microbot opens separately, two grippers open simultaneously, and all four grippers open at the same time (Figure R3.1c), which further illustrates the feasibility and flexibility of the multifocal light field driving strategy.

(b) In our work, the motion of ant microbots is driven by a magnetic field. In magnetic actuation, all magnetic structures within the working range of the magnetic field are subjected to the magnetic field, so simultaneous motion of multiple ant microbots is feasible. As shown in Figure R3.2, three ant microbots are driven by the magnetic field to realize translational and rotational motion at the same time. Since the magnetic materials used by the three ant microbots are the same, and the strength (~ 45 mT) and direction of the magnetic field are almost the same within a small range, their movements in response to the magnetic field are almost synchronized.

(c) Previously, we have verified that the motions of the microbots prepared using the

same magnetic materials are almost synchronized under the same magnetic field. In the usual experiments, to have better magnetic responsiveness of the microbots, the magnetic photoresist used is prepared by mixing commercial photoresist SZ2080 with Fe_3O_4 magnetic nanoparticles surface-modified with oleic acid at a volume ratio of 3:1 (Figure R3.3a). To have variability in the speed of motion among different microbots, we use different magnetic materials to prepare the microbots. For this purpose, we prepare magnetic photoresists with volume ratios of 6:1 and 9:1 (Figure R3.3a). Next, we use a magnet to simultaneously approach ant microbots prepared from different magnetic materials, and the experimental results show that microbots with different magnetic nanoparticle concentrations have different motion speeds under the same magnetic field. The microbot with the highest concentration (3:1) of magnetic nanoparticles moves the fastest, so it could outrun the other two microbots and be assembled individually on a stationary attachment point (Figure R3.3b).

Figure R3.1 A SLM is utilized to modulate multiple beams of light to actuate several grippers open. (a) Schematic diagram of the multifocal light field actuation. (b) Optical diagram of multifocal light fields driving the opening of multiple ant microbot grippers. (c) Optical diagram of multifocal light fields driving multiple grippers opening of a single microbot.

Figure R3.2 The time-sequence optical images of the synchronized motion of ant microbots prepared from the same magnetic material driven by the same magnetic field.

Figure R3.3 Movement of ant microbots prepared from different magnetic materials driven by the same magnetic field. (a) Different magnetic materials are prepared. (b)

The time-sequence optical images of the motion of ant microbots prepared from different magnetic materials driven by the same magnetic field.

List of Revisions:

- ✓ We added the corresponding description on page 10 line 203: *“To manipulate multiple ant microbots at the same time, we can introduce a spatial light modulator to generate multifocal light fields to programmatically control the opening and closing of the mandibles of many ant microbots as well as many claws on a single microbot (Supplementary Figure 16). In addition, by controlling the concentration of magnetic nanoparticles in the magnetic photoresist, multiple ant microbots can achieve synchronized or asynchronized motions driven by the same magnetic field (Supplementary Figure 17-18).”*
- ✓ We added Supplementary Figures 16-18 and the corresponding description in the supplementary materials.

2. Is the ant form optimized in any manner?

Response: We greatly appreciate the reviewer for the constructive comment. In this work, the key to achieving reconfigurable microbot collectives is to control the opening and closing of the microbot mandibles for assembly and disassembly. Therefore, we focus on optimizing the mandible opening ability of the microbot. The opening of the ant microbot mandible is caused by the differential response of different crosslink density layers in the asymmetric bilayer hydrogel joints to changes in temperature, and the opening angle can be managed by controlling the processing parameters of the hydrogel joints. The processing parameters related to the hydrogel joint are shown in Figure R3.4. The parameters affecting the bending angle are the length of the hydrogel joint (l), the width of the low (a) and the high (b) crosslink density layer, as well as the difference in the crosslink density between the low and the high crosslink density layer. It is worth noting that the length (8 μm), width (4 μm), and thickness (4 μm) of the hydrogel joint are always kept constant to harmonize the proportion of the overall structural design. We mainly discuss the optimization of the width of the low (a) and

the high (b) crosslink density layer, as well as the difference in crosslink density between them.

We first discuss the effect of the difference in crosslink density between the low and high crosslink density layers on the mandible opening angle. Since different crosslink densities are realized by different scanning times, we investigated the effect of different scanning times on the opening angle. As shown in Figure R3.5, keeping all other parameters the same, we keep the scanning time of the low crosslink density layer always 1 ms, and the scanning time of the high crosslink density layer is increased from 1 ms to 5 ms. Under stimulation with the same laser power (20 mW), the mandibles open for all parameters except that the joint with the same scanning time exhibits only axial contraction. The opening angle increases and then decreases with increasing scanning time of the high crosslink density layer. Thus, the opening angle of the microbot mandible with a scanning time of 3 ms for the high crosslink density layer is the largest.

Next, we discuss the effect of the width of the low and the high crosslink density layers on the opening angle. As shown in Figure R3.6, keeping other parameters constant. microbots with high crosslink density layer widths ranging from 0.1 to 0.5 of the total joint width are stimulated with the same power of laser, respectively. The mandibles of the microbots with high crosslink density of different widths can all be opened, but the opening angle increases and then decreases with the increasing the width of high crosslink density layer. Thus, the microbot with a high crosslink density layer width ratio of 0.3 has the largest opening angle.

In summary, to maximize the opening angle of the microbot mandible, the width ratio occupied by the high crosslinking density layer in the hydrogel joint is 0.3, and its scanning time should be 3 ms.

Figure R3.4 Schematic diagram of hydrogel joint design parameters.

Figure R3.5 Effect of the scanning time of the high crosslink density layer in the hydrogel joint on the mandible opening angle. All error bars are standard deviations ($n = 3$).

Figure R3.6 Effect of the width of the high crosslink density layer in the hydrogel joint on the mandible opening angle. All error bars are standard deviations ($n = 3$).

List of Revisions:

- ✓ We added the corresponding description on page 7 line 119: “For hydrogel joints, we systematically investigate the effect of the difference and the width ratio of different crosslink density layers on their bending angle (Supplementary Figure 7-8).”
- ✓ We added Supplementary Figures 7-8 and the corresponding description in the supplementary materials.

3. Why only a single attachment point and a single gripper? Ants can grab with mouth and their 6 feet, and can attach at many points along the legs and head. the current

construction only allows chains.

Adding multiple attachment points and grippers would enable tree structures and more rigid connections. Using smooth attachment points would enable pivot points.

Response: We thank the reviewer for this insightful suggestion. As stated by the reviewer, ants can grab with mouths and their six feet, and can attach at many points along the legs and head. Based on this, we design the microbot containing a rectangular body, two claws, and six legs (Figure R3.7a). Each of these legs can be used as an attachment point to assemble (Figure R3.7b), and two microbots can be assembled on one side of the microbot's body without interfering with each other (Figure R3.7c). By further increasing the number of microbots, more complex bifurcated tree-like patterns can be constructed.

The attachment point can be not only a square shape, but also other shapes such as a circle, a triangle, and a semicircle (Figure R3.8). As the reviewer says, the smooth circular attachment point allows for a variety of assembly angles compared to other shape attachment points (Figure R3.9).

Figure R3.7 A microbot with two claws and six legs. (a) Schematic diagram and optical

image of the microbot. **(b)** Optical images of each leg of the microbot used as a gripping point for assembly. **(c)** Optical diagrams of complex bifurcated tree-like patterns assembled from the microbots.

Figure R3.8 Optical images of different shaped attachment points.

Figure R3.9 Optical images of an ant microbot and a circular attachment point are assembled at multiple angles.

List of Revisions:

✓ We added the corresponding description on page 10 line 197: *“In addition to the ant microbot, our assembly strategy is equally applicable to other shaped microbots with multiple claws and gripping points. Thanks to the flexibility of our fabrication approach, we have also designed other shaped microbots and realized more complex assembly forms (Supplementary Figure 13). For example, rectangular-shaped microbots with two claws and six gripping points can be constructed into bifurcated tree-like patterns (Supplementary Figure 14), and disk-shaped microbots with two claws and two gripping points can be assembled into shapes dispersing from the center outward (Supplementary Figure 15).”*

✓ We added Supplementary Figures 13-15 and the corresponding description in the supplementary materials.

“MINOR EDITS”

4. In title, “Ant-inspired” seems too much. The components are ant-shaped, but all the tasks shown could be done by a rectangular block with a gripper and an attachment point.

Response: We are grateful for the valuable comment from the reviewer. The reviewer is correct that the microbot collective reconfiguration strategy proposed in this paper is equally applicable to a rectangular block with a gripper and an attachment point. According to the reviewers’ comments, we perform some new experiments to verify the structure design flexibility of our approach. As shown in Figure R3.10a, we first simplify the body of the microbot into a rectangular shape, which can also be reversibly reconfigured into C, O, V, S, and W shapes by synergistic control of the magnetic and light fields (Figure R3.10b). In addition, we have developed simple disk-shaped microbots with two claws and two gripping points (Figure R3.11a), which can be assembled into shapes dispersing from the center outward (Figure R3.11b).

Although our fabrication and reconfiguration strategies can be applied to microbots of different shapes, this work is inspired by the behavior of ant colonies in nature. Therefore, we choose ant-shaped microbots as the typical structure for our research work. Moreover, the ant-shaped microbot collectives’ behaviors of crossing the gap and transporting heavy micro-objects are also verified, which perfectly matches the behavior of real ant colonies. Therefore, to maintain the source of inspiration and integrity of the study, we want to keep the ant-shaped microbots in this work. We hope and appreciate that the reviewers can understand the preferences and reasons for choosing ant-shaped microbots.

Figure R3.10 Simplified rectangular-shaped microbot. (a) Schematic diagram and optical image of a rectangular-shaped microbot. (b) Optical images of C, J, O, V, S, and W shape assembled from rectangular-shaped microbots.

Figure R3.11 Disk-shaped microbot. (a) Schematic diagram and optical image of a disk-shaped microbot. (b) Optical diagrams of shapes dispersed from the center outward assembled from disk-shaped microbots.

List of Revisions:

✓ We added the corresponding description on page 10 line 197: *“In addition to the ant microbot, our assembly strategy is equally applicable to other shaped microbots with multiple claws and gripping points. Thanks to the flexibility of our fabrication approach, we have also designed other shaped microbots and realized more complex assembly forms (Supplementary Figure 13). For example, rectangular-shaped microbots with two claws and six gripping points can be constructed into bifurcated tree-like patterns (Supplementary Figure 14), and disk-shaped microbots with two claws and two gripping points can be assembled into shapes dispersing from the center*

outward (Supplementary Figure 15).”

✓ We added Supplementary Figures 13-15 and the corresponding description in the supplementary materials.

5. Fig. 1: please show a schematic of the ant, with dimensions. Where are the grasp points?

Response: Thank you for the reviewer’s comment. A schematic of the ant microbot with dimensions has been added to Fig.1 (Figure R3.12).

We apologize for bringing the confusion to the reviewer. As shown in Figure R3.13, Each ant microbot has an L-shaped tail, which is the grasp point.

Figure R3.12 Schematic of the ant microbot with dimensions.

Figure R3.13 Schematic of the gripping point.

List of Revisions:

✓ We revised Figure 1.

6. line 217: What does it mean that “assembled microstructures are destroyed.”? Does this only mean that at least one ant is separated, or does deformation of the structure also count?

Response: We apologize for bringing the confusion to the reviewer. In our work, as shown in Figure R3.14, as soon as the assembled microstructures are deformed during vibration testing, we consider them to be destroyed.

Figure R3.14 Optical comparison images between before and after the assembled microstructures are destroyed. **(a-b)** Optical images of 180° assembly and 90° assembly before destruction. **(c-d)** Optical images of 180° assembly and 90° assembly after destruction. Scale bar, 100 μm .

List of Revisions:

- ✓ We revised the corresponding description on page 12 line 234: *“The green dots represent that the assembled microstructures remain intact and the red cross symbols indicate that the assembled microstructures are destroyed (Supplementary Figure 20).”*
- ✓ We added Supplementary Figure 20 and the corresponding description in the supplementary materials.

7. How is the manipulating magnetic field moved? Is the magnet held under the workspace or above it? Is the magnet moved by a stage or by hand?

Response: We apologize for bringing the confusion to the reviewer. In our work, magnetic manipulations are achieved by changing the position and orientation of a handheld NdFeB magnet relative to the magnetic ant microbots being manipulated. The handheld magnet is slightly above the workspace, and it can provide a strong enough magnetic field strength to manipulate the magnetic ant microbots. For more precise and reliable manipulation, the use of a serial robot arm manipulator with multiple degrees of freedom to carry the actuating magnet and place it around the object being manipulated is desirable.

List of Revisions:

- ✓ We revised the corresponding description on page 9 line 163: *“When a handheld permanent magnet is carefully approached, the ant microbot in the free state will be*

aligned along the direction of the magnetic field and will be able to translate under the magnetic force.”

8. Line 30--31: Shrews don't intertwine, they bite onto the base of the tail of another “a conga line”.

Response: We thank the reviewer for the pointing this out. The sentence has been revised.

List of Revisions:

✓ We revised the corresponding description on page 3 line 31: *“Contrast, more stable collectives are formed by individuals intertwining with each other through mouth or limb deformations as observed in ants^{4,5}, or by one biting on the base of the tail of another in the case of shrews⁶.”*

9. The paper says “claws” many times, but these should all be “mandibles”. Claws are on feet.

Response: We thank the reviewer for pointing this out. The “claw” has been replaced with “mandible” in the manuscript and supplementary material.

10. Line 141, if the jaws only take 20ms to open and close, please run 1000 cycles (which should take around 20 s). This will give a better indication of the longevity of the part.

Response: We thank the reviewer for the comment. To give a better indication of the longevity of the mandibles, we show the opening angle of the mandibles of the ant microbot during 1000 times of switching on/off the illumination in the revised version, and no significant fatigue is observed after repeated stimulation.

Figure R3.15 The microscope images show the initial and the opening state of the ant microbot mandibles at the 200th, 400th, 600th, 800th, and 1000th cycle, respectively.

Figure R3.16 Bending angle of the joints over 1000 cycles of on-off switch of laser with actuation power of 20 mW.

List of Revisions:

- ✓ We revised the corresponding description on page 8 line 142: *“Figure 2g shows the opening angle of the ant microbot during 1000 times of switching on/off the illumination, and no significant fatigue is observed after repeated stimulation, indicating the stable responsiveness (Supplementary Figure 10 and Supplementary Movie 2).”*
- ✓ We revised Figure 2g and the corresponding description on page 26 line 581: *“Bending angle of the joints over 1000 cycles of on-off switch of laser with actuation power of 20 mW.”*
- ✓ We revised Supplementary Figure 10 in the supplementary materials and the corresponding description on page 11 line 86: *“The microscope images show the initial and the opening state of the ant microbot mandibles at the 200th, 400th, 600th, 800th, and 1000th cycle, respectively, and the results demonstrate that the ant microbot mandibles have a promising deformation stability.”*
- ✓ We revised Supplementary Movie 2.

11. Line 153: “the ant microbot rotates sensitively and rapidly excellent controllability”  this is qualitative and not informative. Please instead give the standard measures of rise time, steady-state error, and overshoot.

Response: We thank the reviewer for this helpful suggestion. To test the responsiveness of an ant microbot to an external rotating magnetic field, we place it inside a Helmholtz coil (Figure R3.17). The test is performed by generating a uniform magnetic field in a horizontal plane through the Helmholtz coil. Initially, the ant microbot’s head is oriented toward the positive direction of the X-axis, and when a uniform magnetic field (~45 mT) is applied by the Helmholtz coil along 45° counterclockwise, the ant microbot would rotate counterclockwise following the direction of the magnetic field. Repeating the above steps four times, we finally measure the response time, standard deviation and overshoot are 13.05s, ~0.71°, and 2.35°, respectively.

Figure R3.17 Rotational motion testing. (a) Setup of the rotational motion testing. (b) The time-sequence optical images of rotational motion.

List of Revisions:

- ✓ We revised the corresponding description on page 8 line 153: “*The magnetic force associated with the magnetic field gradient leads to the translation of the ant microbot, and the magnetic torque induces rotational motion of the ant microbot to be aligned along the magnetic field (Figure 2i). We test the ant microbot’s rotational motion capability by using a Helmholtz coil that can generate a uniform magnetic field (~45 mT), and the direction of the magnetic field is changed by 45° each time. The results show that the response time, standard deviation and overshoot are 13.05s, ~0.71°, and 2.35°, respectively. (Figure 2j and Supplementary Figure 12).*”
- ✓ We added Supplementary Figure 12 in the supplementary materials and the

corresponding description.

12. It is unclear how the magnetic field is applied. The pictures show a single permanent magnet. Is this automated or manual? Is there a motor or xy stage to move this magnet? Would it work just as well with a set of electromagnets?

Response: We apologize for bringing the confusion to the reviewer. In our work, magnetic manipulations are achieved by changing the position and orientation of a handheld NdFeB magnet relative to the magnetic ant microbots being manipulated. The handheld magnet is slightly above the workspace, and it can provide a strong enough magnetic field strength to manipulate the magnetic ant microbots. For more precise and reliable manipulation, the use of a serial robot arm manipulator with multiple degrees of freedom to carry the actuating magnet and place it around the object being manipulated is desirable.

In our work, a handheld NdFeB magnet is used for remote magnetic actuation, which can generate a gradient magnetic field to apply a magnetic torque and force to the magnetized ant microbots. Since a Helmholtz coil can produce a uniform magnetic field, and a change in the direction of the uniform field can exert a magnetic moment on magnetic microbots. Another type of Maxwell coil can produce a gradient magnetic field, which can exert a gradient force on magnetic microbots. The magnetic ant microbots can also work well in electromagnets.

List of Revisions:

✓ We revised the corresponding description on page 9 line 163: *“When a handheld permanent magnet is carefully approached, the ant microbot in the free state will be aligned along the direction of the magnetic field and will be able to translate under the magnetic force.”*

13. Fig. 3, which ants are magnetic? (Are all of them?)

Response: We apologize for bringing the confusion to the reviewer. In Fig. 3, we show

the assembly of two ant microbots at 90° and 180° under the coordinated control of magnetic and light fields, as well as the assembly of multiple ant microbots into different morphologies. All ant microbot are magnetic, where the first one is fixed to the substrate and the others are assembled sequentially.

List of Revisions:

✓ We revised the corresponding description on page 28 line 588: *“Figure 3. Reversible, multiple, and robust assembly of magnetic ant microbots.”*

14. Fig. 3 e and f, “Frequence” -> “Frequency”

Response: We apologize for the spelling mistake, and we have corrected it in our revised manuscript.

List of Revisions:

✓ We revised the corresponding description in Figure 3e and f.

“TINY EDITS” (e.g. typos)

15. Figure 1C is a compiled (“fake”) photo of three ants rolling a gooseberry fruit. Please replace with an actual photo of real ants.

<https://depositphotos.com/photo/ants-carrying-food-together-teamwork-concept-162201840.html>

Response: We thank the reviewer for pointing this out. The photo in Figure 1c has been replaced by an actual photo of real ants (Figure R3.18).

Figure R3.18 Actual photo of real ants working together to carry.

List of Revisions:

✓ We revised Figure 1c.

16. Line 13 “individual. However, current various weak force induced microbot collectives still face the” is too many adjectives. Link words modified with a dash: “force-induced”? What is a “weak force”?

Response: We thank the reviewer for the comment. The sentence has been corrected. Researchers have developed microbot collectives capable of reconfiguring their morphology in response to external magnetic field¹, light², electric field³, and ultrasound⁴, enabling a diversity of assembly patterns. However, the assembled shapes require continuous input of external stimuli to maintain and can be easily disrupted by external disturbances. In our work, we call them weak force-induced microbot collectives. In contrast, the microbot collectives assembled using mechanical structures in our research work are more stable against external disturbances.

References:

1. Yu, J., Wang, B., Du, X., Wang, Q. & Zhang, L. Ultra-extensible ribbon-like magnetic microswarm. *Nat. Commun.* 9, 3260 (2018).
2. Chen, M. et al. Programmable dynamic shapes with a swarm of light-powered colloidal motors. *Angew. Chem. Int. Ed.* 60, 16674-16679 (2021).
3. Hernández-Navarro, S., Tierno, P., Farrera, J. A., Ignés-Mullol, J. & Sagués, F. Reconfigurable swarms of nematic colloids controlled by photoactivated surface patterns. *Angew. Chem. Int. Ed.* 53, 10696-10700 (2014).
4. Tang, S. et al. Structure-dependent optical modulation of propulsion and collective behavior of acoustic/light-driven hybrid microbowls. *Adv. Funct. Mater.* 29, 1809003 (2019).

List of Revisions:

✓ We revised the corresponding description on page 1 line 13: *“However, various weak force-induced microbot collectives maintained by magnetic, light, and electric field still face the challenges such as unstable connections, the need for a continuous*

external stimuli source, and imprecise individual control.”

17. Line 15: “imprecisely individual control” -> “imprecise individual control”

Response: We apologize for this grammar mistake, and we have corrected it in our revised manuscript.

List of Revisions:

✓ We revised the corresponding description on page 1 line 13: *“However, various weak force-induced microbot collectives maintained by magnetic, light, and electric field still face the challenges such as unstable connections, the need for a continuous external stimuli source, and imprecise individual control.”*

18. Line 21 “to cross a large gap” is imprecise. Better is “to cross a one-body-length gap”

Response: We thank the reviewer for this suggestion, the “to cross a large gap” has been replaced by “to cross a one-body-length gap”.

List of Revisions:

✓ We revised the corresponding description on page 1 line 21: *“Moreover, we demonstrate the ability of assembled microbots to cross a one-body-length gap and their adaptive capability to move through a constriction and transport microcargo.”*

✓ We revised the corresponding description on page 5 line 71: *“We demonstrate the ability of assembled ant microbots to traverse a one-body-length gap.”*

✓ We revised the corresponding description on page 17 line 357: *“Afterwards, we present the ability of the assembled ant microbots to traverse a one-body-length gap.”*

19. Line 22: “transport a microcargo.” -> “transport microcargo.”

Response: We apologize for the grammar mistake. The sentence has been corrected.

List of Revisions:

✓ We revised the corresponding description on page 1 line 21: *“Moreover, we demonstrate the ability of assembled microbots to cross a one-body-length gap and their adaptive capability to move through a constriction and transport microcargo.”*

✓ We revised the corresponding description on page 25 line 565: *“Three ant microbots assembled in C shape mimic ants transporting cargo.”*

20. Line 22 “the field of clustered microbots functions,” no idea what this is. maybe “the abilities of microbots,”?

Response: We apologize for bringing the confusion to the reviewer. The sentence has been revised.

List of Revisions:

✓ We revised the corresponding description on page 1 line 22: *“Our strategy will broaden the abilities of clustered microbots, including gap traversal, micro-object manipulation, and drug delivery.”*

21. Line 28 “One type, in which there is no direct contact between individuals” -> “In the first category there is no direct contact between individuals”

Response: We thank the reviewer for the comment. The sentence has been revised.

List of Revisions:

✓ We revised the corresponding description on page 3 line 29: *“In the first category there is no direct contact between individuals, as in the case of bees¹, fishes², and birds³, causing these collectives tend to be unstable and vulnerable to disruption.”*

22. Line 30 “contrast, the other, more stable collectives” -> “contrast, more stable collectives”

Response: We thank the reviewer for pointing this out. The sentence has been revised.

List of Revisions:

✓ We revised the corresponding description on page 3 line 31: “*Contrast, more stable collectives are formed by individuals intertwining with each other through mouth or limb deformations as observed in ants^{4,5}, or by one biting on the base of the tail of another in the case of shrews⁶.*”

23. Line 31 “These collectives formed” -> “These collectives are formed”

Response: We thank the reviewer for the comment. The sentence has been revised.

List of Revisions:

✓ We revised the corresponding description on page 3 line 33: “*These collectives are formed by individuals locking onto each other can resist interference and always remain as an integrated whole.*”

24. Line 32 “locking into each” -> “locking onto each”

Response: We apologize for the grammar mistake. The sentence has been corrected.

List of Revisions:

✓ We revised the corresponding description on page 3 line 33: “*These collectives are formed by individuals locking onto each other can resist interference and always remain as an integrated whole.*”

25. Line 38-39 “individually actuated for specified unit” -> “individually actuated”.

Response: We thank the reviewer for the comment. The sentence has been revised.

List of Revisions:

✓ We revised the corresponding description on page 3 line 37: “*Although these assembly strategies can reconfigure the morphology of micro/nano swarms to perform certain collective behaviors¹⁴⁻¹⁶, the monomers are poorly controllable and cannot be selectively and individually actuated.*”

26. Line 46 “controlled and steady manner.” -> “controlled and stable manner.”

Response: We thank the reviewer for pointing this out. The sentence has been revised.

List of Revisions:

✓ We revised the corresponding description on page 3 line 45: *“Inspired by this behavior, using a deformable microbot as the basic unit is a promising method to construct microbot collectives that can be assembled and disassembled in a controlled and stable manner.”*

27. Line 50 “predefined director field orientations” what is this? “predefined field orientation directions”?

Response: We apologize for bringing the confusion to the reviewer. In the reference, the authors used liquid crystal elastomers (LCEs) to fabricate temperature responsive voxels. In general, liquid crystal molecules require friction to accomplish their preorientation, often referred to as “predefined orientations”. When an external temperature or light stimulus is applied, the oriented arrangement of liquid crystal molecules will turn to a non-oriented random arrangement, which in turn leads to a complex deformation of the macroscopic structure.

Reference:

Guo, Y., Zhang, J., Hu, W., Khan, M. T. A. & Sitti, M. Shape-programmable liquid crystal elastomer structures with arbitrary three-dimensional director fields and geometries. *Nat. Commun.* **12**, 5936 (2021).

28. Line 110 “and instantaneous contraction can be observed”  8 ms is not instantaneous. -> “and full contraction in 8 ms is observed”

Response: We thank the reviewer for pointing this out. The sentence has been revised.

List of Revisions:

✓ We revised the corresponding description on page 6 line 109: *“In this work, in*

order to quantitatively investigate the expansion and contraction properties of this hydrogel, micro circular plates with a diameter of 20 μm and a thickness of 3 μm are fabricated and measured and full contraction in 8 ms is observed when the microplate is illuminated.”

29. pg 6, what are the units for SRT?

Response: We apologize for bringing the confusion to the reviewer. In our work, we define the scanning repeat time as the SRT, where the scanning repeat time is the dwell time of the laser focal spot at each scanning point position, so the unit of SRT is ms. We have revised the relevant descriptions in the manuscript.

List of Revisions:

✓ We revised the corresponding description on page 7 line 114: *“In this study, taking advantage of the programmable fabrication capability^{44,45}, we keep the processing laser power and scanning point step constant at 33 mW and 320 nm, respectively, while choosing scanning repeat time (SRT) as the processing variable to control the shrinking ratio of the hydrogel, and the maximum shrinking ratios for SRT = 1 ms and SRT = 3 ms are ~0.39 and ~0.1, respectively (Supplementary Figure 6).”*

✓ We revised the corresponding description on page 7 line 120: *“Based on these experimental results, we design a micro asymmetrical double-layered joint (8 μm in length and 4 μm in width) to achieve bending deformation, where the width ratio of the SRT = 1 ms region to the total width is 0.7 and the remaining part of SRT = 3 ms.”*

✓ We revised the corresponding description on page 7 line 59 in the supplementary material: *“In order to realize the bending deformation of the joint, it is designed so that the width ratio of the SRT = 1 ms region to the total width is 0.7 and the remaining part of SRT = 3 ms.”*

✓ We revised the corresponding description on page 24 line 168 in the supplementary material: *“In this way, we obtain thermal shrinkage ratio $\alpha_1 = 0.39$ and $\alpha_2 = 0.1$ for the hydrogel microplates with SRT of 1 ms and SRT of 3 ms, respectively.”*

✓ We revised the corresponding description in Supplementary Figure 6b in the

supplementary material.

30. line 125: “temperature of joint is” -> “temperature of the joint is”

Response: We apologize for the grammar mistake. The sentence has been corrected.

List of Revisions:

✓ We revised the corresponding description on page 7 line 128: *“Specifically, as soon as the temperature of the joint is higher than the LCST of the hydrogel via surface incorporated photo absorbers, it immediately bends toward the side of the fewer SRT due to the larger shrinking ratio of the fewer SRT than the more SRT (Supplementary Figure 6).”*

31. line 129 “actuated by the joints to become the opening state.” -> “actuated by the joints to open.”

Response: We thank the reviewer for the comment. The sentence has been revised.

List of Revisions:

✓ We revised the corresponding description on page 7 line 130: *“In this way, the mandibles initially close together, and once the light field perpendicular to the surface of the head is applied, the mandibles will be actuated by the joints to open.”*

32. line 157 “As the schematics shown in Figure 3a,” -> “As the schematics show in Figure 3a,”

Response: We apologize for the grammar mistake. The sentence has been corrected.

List of Revisions:

✓ We revised the corresponding description on page 8 line 162: *“As the schematics show in Figure 3a, there are two ant microbots, and one of them is fixed to the substrate while the other is free to move.”*

33. line 184 “number of the two assemble modes,” -> “number of the two assembly modes,”

Response: We thank the reviewer for pointing this out. The sentence has been revised.

List of Revisions:

✓ We revised the corresponding description on page 10 line 189: “*As the demonstrations illustrated in Figure 3c, by changing the sequence and number of the two assembly modes, numerous configurations of ant microbot can be realized.*”

34. line 192 “Besides that, the” -> “The”

Response: We thank the reviewer for the comment. The sentence has been revised.

List of Revisions:

✓ We revised the corresponding description on page 11 line 210: “*The controllable and selective assembly strategy can also enable the assembly between multiple non-magnetic material units with the assistance of a magnetic unit (Supplementary Movie 5-6).*”

35. Line 207 “configurations that close resemblance to the tetrominoes” -> “configurations that closely resemble the tetrominoes”

Response: We thank the reviewer for this suggestion. The sentence has been revised.

List of Revisions:

✓ We revised the corresponding description on page 11 line 223: “*To illustrate the dynamic control capability of the magnetic unit to manipulate different non-magnetic material units in a single coding sequence, we pattern different material units into various configurations that closely resemble the tetrominoes in the popular game Tetris (Supplementary Figure 19).*”

36. Line 226: add the ant height and length so the reader can see the correlation.

Response: We thank the reviewer for the comment. To show the correlation in size between the ant and the gap, we have added the height and length of the ant.

List of Revisions:

✓ We revised the corresponding description on page 12 line 244: *“In detail, the ant and the gap lengths are 97 μm and 120 μm , respectively, while the ant and the gap heights are both 20 μm (Figure 4a).”*

37. Line 229: “the center of gravity (CG) of the microbot is located in the middle pair of legs” -> “the center of gravity (CG) of the microbot is located above the middle pair of legs”

Response: We apologize for the grammar mistake. The sentence has been corrected.

List of Revisions:

✓ We revised the corresponding description on page 12 line 246: *“For a single robot performing forward locomotion and attempting to cross the gap, most failures occur after the middle pair of legs fall into the gap because the center of gravity (CG) of the microbot is located above the middle pair of legs.”*

38. Line 252 “It is worth noting that the width” -> “The width”

Response: We thank the reviewer for the comment. The sentence has been revised.

List of Revisions:

✓ We revised the corresponding description on page 13 line 276: *“The width of the entrance and exit of the maze allow only I-shaped three-unit microbot to pass through and the entrance and exit of the maze have a gripping point to facilitate the switching of the assembly pattern.”*

39. Lines 281 and 283 “drugs carrier” -> “drug carrier”

Response: We thank the reviewer for pointing this out. The sentence has been revised.

List of Revisions:

✓ We revised the corresponding description on page 15 line 304: “*we manipulate the three-unit microbot to transport the cubic drug carrier (20 μm in length) from the entrance of the maze to the experimental group by coordinating the magnetic and light fields, where the drug carrier is prepared by immersing a cubic hydrogel block processed by TPP in doxorubicin hydrochloride (DOX) solution (5 mg mL^{-1}) for 1 h to load drugs onto the carrier. Figure 5b is the time-lapse images of drug carrier delivery, and the images from each location have been stitched together to demonstrate the full path.*”

40. Line 285 “gradually increase over” -> “gradually increases over”

Response: We apologize for the grammar mistake. The sentence has been corrected.

List of Revisions:

✓ We revised the corresponding description on page 15 line 310: “*After the carrier is delivered to the targeted position, the extent and intensity of the red fluorescence gradually increases over time and need approximately 20 min to fill the entire area of the experimental group, indicating that the drugs had diffused rapidly from the carrier (Supplementary Figure 22).*”

41. Fig. 5, for (n=3), please make the n italic.

Response: We thank the reviewer for the comment. The font of n has been revised.

VIDEO:

42. Supplementary material: the same icon for a laser is used in Fig. 1 a and b. Is the same wavelength used? (also Fig. 2a)

Response: We apologize for bringing the confusion to the reviewer. In our work, the fabrication and actuation of the ant microbot are realized in the same femtosecond (Fs) laser system, so the wavelengths of the lasers used are all the same. Specifically, the

central wavelength of the laser is 800 nm with a pulse width of 75 fs and a repetition rate of 80 MHz. We have added some descriptions to Supplementary Figure 1 in the supplementary material to help readers better understand.

List of Revisions:

✓ We added the corresponding description on page 2 line 25: “*The wavelength of the laser used to fabricate and actuate the ant microbot is 800 nm.*” in Supplementary Figure 1 in the supplementary material.

43. Fig. 10 is hard to see the difference in colorings. Are the orientations of the magnetic material the same in each ant? If so, in Fig. 10 C, column2, how are the three magnetic photoresist ants not all in the same orientation?

Response: We apologize for bringing the confusion to the reviewer. The colors have been re-added to easily show the difference of the materials. In our work, the magnetic materials are prepared by directly mixing homemade Fe₃O₄ magnetic particles modified with Oleic acid with commercially available photoresists SZ2080, and each magnetic ant microbot is fabricated by the same processing method, so the orientations of the magnetic materials are the same in each magnetic ant microbot.

As shown in Figure R3.19, to pattern ten individual units containing four different material units into a rectangular configuration, a photoresist unit is initially immobilized on a glass substrate. A PETA gel unit and a magnetic photoresist unit are transferred to the vicinity of the photoresist unit through a glass microneedle. Afterwards, the magnetic photoresist unit guides the PETA gel unit to assemble with the photoresist unit at 90° under the coordinated control of magnetic and light fields (Figure R3.19a-d). Next, a PEGDA gel unit is transferred to the vicinity of the assembled microstructure using a glass microneedle. The same control is utilized to allow the magnetic unit to guide the PEGDA gel unit to assemble with the PETA gel unit at 180° (Figure R3.19e-g). In this way, the on-demand assembly between four different material units is realized. It is worth noting that since the assembled units are sufficiently stable, they will remain in their assembled state if they are only subjected

to a magnetic field, which is the reason that the new magnetic units can guide the non-magnetic units to accomplish further normal assembly. Repeating the previous steps, ten individual units containing four different materials can be assembled into a rectangular configuration (Figure R3.19h-x), and the three magnetic photoresist units are not all in the same orientation.

Figure R3.19 The time-sequence optical images of the process of assembling ten individual units containing four different materials into a rectangular configuration. Scale bar, 100 μm .

List of Revisions:

✓ We revised Supplementary Figure 19 in the supplementary material.

44. Fig. 11, please add time stamps to the images.

Response: We thank the reviewer for this suggestion. The time stamps have been added to the Supplementary Figure 21 (Figure R3.20).

Figure R3.20 Maze navigation and cargo delivery.

List of Revisions:

✓ We revised Supplementary Figure 21 in the supplementary material.

Article File PDF (1393KB) Source File (DOCX) 4207KB

Supplementary Information PDF (1570KB) Source File (DOCX) 4769KB

45. Video - Supplementary Movie 1 Video (5474KB) Source File (MP4) 5474KB -- tiny 'ant' that opens and closes jaws. Scale bar at bottom right

46. Video - Supplementary Movie 2 Video (4142KB) Source File (MP4) 4142KB -- tiny 'ant' (same as video 1?) that opens and closes jaws 50x. Scale bar at bottom right

Response: We apologize for bringing the confusion to the reviewer. Supplementary Movie 1 shows the dynamic deformation process of the ant microbot mandibles opening and closing. And Supplementary Movie 2 shows the opening angle of the mandibles of the ant microbot during 1000 times of switching on/off the illumination, and no significant fatigue is observed after repeated stimulation, indicating the stable responsiveness.

47. Video - Supplementary Movie 3 Video (4122KB) Source File (MP4) 4122KB -- one ant approaches another and grasps it. Unclear why the first ant does not move. Also hard to tell when light is on -- better to always have an annotation that says light is off. Also hard to tell which movement is due to the camera moving and which is due to tiny robot moving. Adding a stationary landmark would help.

Response: We apologize for bringing the confusion to the reviewer. Both ant microbots are magnetic and can be actuated by a magnetic field. But the first ant is not poked off using a glass micro-needle based on a micro-manipulation system, whereas the second ant is poked off and is free to move, so the first ant will not move under a magnetic field. It is also possible to temporarily hold onto a fixed microstructure by deformation of the mandibles of the first ant to avoid being subjected to a magnetic field.

We thank the reviewer for these suggestions. Annotations have been added to the Supplementary Movie 3, 4, 5, and 6 to make it easy for the readers to tell when the light is on and when it is off. Since the first ant microbot is fixed to the substrate, it can be used as a stationary landmark to mark it easy for the readers to tell the movement is camera moving or ant microbot moving. When the first ant microbot and the other ant microbot move, this means it is the camera moving. When the first ant microbot doesn't move and the other ant microbot moves, this means it is the ant microbot moving.

List of Revisions:

✓ We revised Supplementary Movies 3, 4, 5, and 6.

48. Video - Supplementary Movie 4 Video (6125KB) Source File (MP4) 6125KB -- ant

approaches at 90 degrees. What is the ant grasping? Is the B field the orientation of the field, or the orientation of the gradient, or both?

Response: We apologize for bringing the confusion to the reviewer. As shown in Figure R3.21, Each microbot has a mandible and an L-shaped tail. The former ant microbot is anchored to the substrate while the latter is free to move (Figure R3.21a). Then, a magnetic field actuates the free unit to approach the fixed unit. When the latter mandible contacts with the former tail, a light field is introduced to drive the latter mandible open. After that, the latter continues to move forward driven by the magnetic field so that the latter mandible can encircle the former's tail (Figure R3.21b). Finally, after turning off the light field the latter mandible tightly grasps the former tail in order to realize a 180° assembly of the two ant microbots (Figure R3.21c). Similarly, using this strategy the two ant microbots can also achieve 90° assembly on-demand.

In our work, a gradient magnetic field is generated by a magnet that applies a magnetic torque and force to the magnetized ant microbots. Therefore, the B field is the orientation of the magnetic field gradient.

Figure R3.21 Schematics of two ant microbots achieving the 180° assembly process.

49. Video - Supplementary Movie 5 Video (5343KB) Source File (MP4) 5343KB -- two-chain ant approaches 3rd ant from back. What is the radius of the light beam? Where is the focus? (It can make a solitary ant open or close its jaws.)

Response: Thanks to the reviewer for the good questions. In our work, a $5\times$ ($NA = 0.15$)

objective lens is used. For a focused Gaussian beam, the spot radius is $0.61\lambda/\text{NA}$ (where λ and NA are the wavelength and numerical aperture of the objective lens, respectively). Therefore, in theory, the radius of the spot at the focus is $\sim 3.25\ \mu\text{m}$. Correspondingly, the actual radius of the light beam is measured to be $\sim 3.3\ \mu\text{m}$ (Figure R3.22a), which agrees well with the simulation results.

As shown in Figure R3.22b, the light beam is focused on the head which is close to the mandible and has silver nanoparticles (Ag NPs), where the Ag NPs are used for photothermal conversion. When Ag NPs are irradiated by NIR light, strong photothermal conversion induces the temperature increase in hydrogel joints to trigger mandible opening.

Figure R3.22 (a) Optical images of a focused spot under a $5\times$ objective lens. (b) Schematic of spot focusing position. Scale bar, $10\ \mu\text{m}$.

List of Revisions:

✓ We added the corresponding description on page 26 line 573: *“The laser is used to trigger the ant mandibles to open, and the radius of the focused spot under the $5\times$ objective lens is approximately $3.3\ \mu\text{m}$.”*

50. Video - Supplementary Movie 6 Video (5126KB) Source File (MP4) 5126KB -- two-chain ant approaches 3rd ant from 90-deg.

51. Video - Supplementary Movie 7 Video (4566KB) Source File (MP4) 4566KB -- two-chain ant approaches and connects to 3rd ant from back, then back ant separates and moves away.

52. Video - Supplementary Movie 8 Video (14223KB) Source File (MP4) 14223KB --

“u”-shaped 3 ant formation is placed on a shake table? (how does that work?)
Misspelling “Untabe” -> “Unstable” at 360 micrometer amplitude.

Response: Many thanks to the reviewer’s valuable comments. To test the stability of the assembly, the assembled structure is placed on a horizontal vibrating platform. Note that the first unit is anchored to the substrate to vibrate with the platform, while the remaining two units are assembled by our proposed strategy. The vibration frequency and amplitude are modulated by a function waveform generator (Figure R3.23).

We apologize for the misspelling of the word. The word has been corrected.

Figure R3.23 Schematic of the vibration test setup.

53. Video - Supplementary Movie 9 Video (7911KB) Source File (MP4) 7911KB -- gap traversal. Please show the magnetic field. What is the height of the gap? A schematic showing a side view would help immensely. (show different illustration for all 3 tests, and show the failure case.

Response: Thanks for the reviewer’s careful review and nice advice. The magnetic field and schematics of the side views have been shown in Figure 4. The height of the gap is 20 μm .

List of Revisions:

✓ We revised Figure 4.

54. Video - Supplementary Movie 10 Video (19197KB) Source File (MP4) 19197KB -
- three ants reconfigure into a “u” shape and move a square object.

55. This last video is composed of many takes, and the square box jumps around, often

from a bad position to a “good” position in the next take. How many trials were conducted and spliced together to make this video? The video at present is unacceptable and misleading. At minimum, the trial number for each video segment needs to be listed.

Response: We thank the reviewer for the comment. The Supplementary Movie 10 contains three main segments. The first segment shows the microbot collectives entering the maze in an I-shape and switching to a C-shape. The second segment shows the C-shaped microbot collectives transporting the cargo. The last segment shows the microbot collectives switching to an I-shape and leaving the maze. Due to the complexity of the whole process, we conduct three, one, and three times in each of these three segments and splice together to make the Supplementary Movie 10. We illustrate the number of experiments conducted for each of the three segments in Supplementary Figure 21.

List of Revisions:

✓ We added the corresponding description in Supplementary Figure 21: “(a) We conduct this segment three times. (b) We conduct this segment one time. (c) We conduct this segment three times.”

56. A schematic showing a side view would help. Which robot(s) are magnetic?

Response: We sincerely appreciate the reviewer’s kind advice. Following the reviewer’s suggestion, the related schematics of side view have been added to the Supplementary Figure 21 (Figure R3.24), in which all three ant microbots are magnetic. We hope this will help reviewers and readers understand better.

Figure R3.24 Maze navigation and cargo delivery.

List of Revisions:

- ✓ We revised Supplementary Figure 21.

REVIEWER COMMENTS

Reviewer #1 (Remarks to the Author):

Comparison with Existing IVF Technologies: The manuscript references in vitro fertilization (IVF) applications to justify its approach. Given the advancements in automated IVF technologies, including microfluidics and automated micromanipulation, it is essential for this study to provide a comparative analysis with these established methods. Notably, the use of optical tweezers in IVF raises questions about the unique contributions of the project. If optical tweezers can effectively manipulate cells, the need for the proposed magnetic control system as a wireless actuation approach should be clearly justified.

Clarification on Optical Fiber Utilization: The explanation regarding the use of optical fibers in in vivo environments remains unclear. If the optical fibers can access target sites directly, the rationale for employing an additional system needs to be more convincingly detailed.

Experimental Validation of New Figures: The addition of Figure Fig. R1.2 re-emphasizes certain aspects of the manuscript. It is crucial that the claims and conceptual illustrations in this figure are supported by experimental evidence, particularly since they appear to play a significant role in justifying the project's motivation. This is closely related to my previous comment 4. Since Fig. 5 is still there, please do this experiment.

Foundation of Manuscript's Motivation: The manuscript's motivation, outlined in lines 56-57, appears to be based on the premise that "microbot collectives based on these intelligent units and their applications have not been validated." This foundational statement seems rather tenuous. It would strengthen the manuscript if you could more explicitly articulate the rationale behind the chosen experimental approach and its significance in advancing the field.

Reviewer #2 (Remarks to the Author):

Thank you for taking the time to review my comments - I especially appreciate seeing the new robot shapes. Overall I'm satisfied with this version of the paper.

Reviewer #3 (Remarks to the Author):

Review of "Ant-inspired reconfigurable microbot collectives"

Second review. This is a fun contribution. The ability to lock individual components in at 0/90/180/270 degree angles to make rigid attachments using a default closed gripper is clever. The experiments are interesting and the results are easy to understand. I didn't review the supplemental material carefully. The "Tiny edits" are suggestions to make the paper easier to read.

"BIG ISSUES" In order of importance

Many images in the supplementary material are more exciting than the revised paper. I really like fig 14 and fig. 15 and fig. 16 in the supplemental material.

"MINOR EDITS"

Fig. 1 a, why is the right column labeled as 'reversible', but the 180 degree isn't reversible. Aren't all assemblies reversible?

Fig. 1 b, the gap is not shown to scale. 3 ants could not bridge this large gap.

Fig. 2 g claims to show 1000 cycles, but only 9 cycles are shown. Was the wrong image uploaded?

Fig. 4 c,d,e the gap is not shown to scale. A three-ant assembly could not bridge the gap shown in fig. 4e. Please regenerate with a scale image.

Fig. 4 b, please add a dotted line labelled 'b_1' to show the length of one robot.

Fig. 5.d. "Contol" in legend should be "Control".

Fig. 5.c would benefit from a labelled color bar or legend for the green and black color.

"TINY EDITS" (e.g. typos)

"13 individual. However, various weak force-induced microbot collectives maintained by magnetic, 14 light, and electric field still face the challenges such as unstable connections, the need for a 15 continuous external stimuli source, and imprecise individual control. Here, we construct magnetic" ->

"13 individual. However, various force-induced microbot collectives maintained by weak magnetic, 14 light, or electric fields still face challenges such as unstable connections, the need for a 15 continuous external stimuli source, and imprecise individual control. Here, we construct magnetic"

"30 bees¹, fishes², and birds³, causing these collectives tend to be unstable and vulnerable to disruption.

31 Contrast, more stable collectives are formed by individuals intertwining with each other through" 

"30 bees¹, fishes², and birds³, causing these collectives to be unstable and vulnerable to disruption.

31 In contrast, more stable collectives are formed by individuals intertwining with each other through

"33 in the case of shrews⁶. These collectives are formed by individuals locking onto each other can" ->

"33 in the case of shrews⁶. These collectives are formed by individuals locking onto each other. They

can"

This one is more confusing. Perhaps:

"51 typical examples, Sitti et al. created cubic liquid crystal elastomer voxels with predefined director field orientations and stitched these voxels into lines, grids, and skeletal structures for" ->

"51 typical examples, Sitti et al. created cubic liquid crystal elastomer voxels with predefined field orientations directions and stitched these voxels into lines, grids, and skeletal structures for

"62 process32-36, and then the silver nanoparticles (Ag NPs) are deposited onto the head of the ant" ->

"62 process32-36, and then silver nanoparticles (Ag NPs) are deposited onto the head of the ant"

"64 Ag NPs, the ant mandibles could open immediately when the laser is focused on the ant head. " ->

"64 Ag NPs, the ant mandibles open immediately when the laser is focused on the ant head."

109 & 133: "In order to" -> "To".

125 'It should be noted that the" -> "The

184 "the latter mandibles could encircle" -> "the latter mandibles encircle"

195 "into the J shape" -> "into a J shape".

238 "the stability of C shape is larger than that of I shape" ->"the stability of the C shape is larger than that of the I shape"

"269 The ant colonies in nature contain many individuals, which can collaborate to accomplish various tasks. In contrast, the number of individuals in the collectives formed by our proposed assembly strategy is difficult to be as large as that of ant colonies." ->

"269 Ant colonies in nature contain many individuals, which can collaborate to accomplish various tasks. In contrast, it is difficult to use our assembly strategy for large numbers of individuals."

277 "allow only I-shaped" -> "allow only an I-shaped"

309 "Figure 5b is the time-lapse images of drug carrier delivery -> " Figure 5b shows time-lapse images of drug delivery"

311 "increases over time and need" -> "increases over time and needs"

321 "Microbot collectives have a superior ability to accomplish complex tasks compared to individual microbot." -> "Microbot collectives have a superior ability to accomplish complex tasks compared to

individual microbots."

325 "the ability to traverse gap" -> "the ability to traverse gaps"

339 "In our proposed assembly strategy, gripping point" -> "In our proposed assembly strategy, a gripping point"

Reply to the Reviewers' and editor's comments

Thanks a lot for the comments. The manuscript has been carefully modified according to the comments. The revisions were marked in red as shown below. We appreciate the valuable comments and suggestions from the reviewers and editor. The point-to-point answers to the comments of the reviewers and editor are listed as follows. There is also a change list in the end. (The numbers of Page and line in "Revision list" correspond to "Manuscript with detailed changes.pdf" and "SI with detailed changes.pdf")

Reviewer #1 (Remarks to the Author):

1. Comparison with Existing IVF Technologies: The manuscript references in vitro fertilization (IVF) applications to justify its approach. Given the advancements in automated IVF technologies, including microfluidics and automated micromanipulation, it is essential for this study to provide a comparative analysis with these established methods. Notably, the use of optical tweezers in IVF raises questions about the unique contributions of the project. If optical tweezers can effectively manipulate cells, the need for the proposed magnetic control system as a wireless actuation approach should be clearly justified.

Response: We thank a lot for the reviewer’s comment. According to your advice, the approach proposed by our work is systemically compared with other technologies. The details were shown in the Table R1.

Table R1 Comparison of techniques for manipulating cells

Techniques for manipulating cells	Direct contact	Group manipulation	Manipulating cell size	Arbitrariness of manipulation	Damage
Automated micromanipulation ^{1,2}	Yes	No	Depend on the size of the micropipettes	Yes (3D)	Yes (Mechanical damage)
Microfluidics ^{3,4,5}	No	Yes	Depend on the size of the microchannels	No (Depend on the design of the microfluidics)	NA
Optical tweezers ^{6,7}	No	Yes	10 nm~10 μm	Yes (3D)	Yes (Optical damage)
This work	Yes	Yes	5~100 μm	Yes (3D)	No

Specific comparisons are listed as follows:

In vitro fertilization (IVF) is a complex series of processes including the selection of motile sperm, retrieval and processing of oocytes, insemination, in vitro embryo culture, and cryopreservation. Here, the prospective application of our proposed method is applicable to carrying sperm and oocytes together, followed by the natural penetration of the sperm into the oocyte and thus fertilization occurs.

(1) We consider your proposed automated micromanipulation system as an automated microinjection system. It is a very complex system that includes machine vision, nonvision sensors and user interface (input), cell modeling, piercing mechanisms and injection control loop (control), cell holder and manipulator, and microinjection (output). Since insertion of micropipette into the oocyte is a mechanically invasive technique, there is a risk of oocyte lysis, or degeneration. In addition, the size of manipulable cells is limited by the diameter of the micropipette and it is difficult to manipulate in bulk.

(2) Microfluidics technology provides an alternative to almost every single step in the process of VIF, including selection of motile sperm, automated processing of oocytes, in vitro follicle growth, embryo culture and cryopreservation. However, limited by the microfluidic channel, the size of the cells cannot exceed the width of the microchannel and the cells cannot be manipulated to arbitrary positions. In addition, most of the microfluidic devices fabricated for IVF so far are made of PDMS. This type of transparent and gas-permeable elastomer potentially has negative effects for cell studies at the microscale, such as deformation, molecular absorption, hydrophobic recovery, evaporation of culture medium, and leaching of uncross-linked oligomers⁵.

(3) Optical tweezers, which are three-dimensional optical traps formed through tightly focusing a laser beam with high numerical aperture objective lens, are a non-invasive method to manipulate objects in solution with sizes ranging from tens of nanometers to a few micrometers. While offering versatility and high precision, they also have some drawbacks. The first point is that optical interference affecting the intensity and intensity distribution can degrade the performance of optical tweezers since the optical trap stiffness depends on the gradient of the optical field. And although multiple optical traps can be created by polarization splitting or rapid beam scanning to control multiple objects simultaneously, disturbances in the optics and beam steering optics can lead to spurious signals. The second point is that the high intensity at the focus of the trapping laser that forms the optical trap, typically 10^9 - 10^{12} W cm⁻¹, results in local heating, and the local heating can affect enzymatic activity and change the local viscosity of the

medium to cause optical damage. Compared to optical tweezers, our proposed coordinated control strategy of magnetic and optical fields can effectively circumvent the above problems. The microbot collectives assembled by different number of magnetic units can manipulate objects from a few micrometers to tens of micrometers under the control of a magnetic field, and they have a stronger anti-interference ability. In addition, the method does not suffer from localized heating and optical damage problems.

Currently, researchers have proposed using artificial magnetically controlled microbots to manipulate sperms^{8,9}. As a typical example, Oliver G. Schmidt et al. showed the capture and transportation of immobile, but otherwise functional sperms to the oocyte by coupling them to artificial helical micromotors that can be actuated by rotating magnetic fields⁹. The above studies have verified that microbots provide new tools for in vitro fertilization, although there are many challenges in practical applications. Back to our work, due to the limited experimental conditions in biology, it is difficult for us to experimentally verify the process of in vitro fertilization. More importantly, we would like to reiterate that the main contribution of this work is to propose a stable and reliable microbot collective reconfiguration scheme based on mechanical assembly and disassembly via optical-magnetic cooperative control. In summary, according to the editor's suggestion, we deemphasized the part on cell manipulation in the revised manuscript.

References:

1. Chow, Y. T. et al. A high-throughput automated microinjection system for human cells with small size. *IEEE/ASME Transactions on Mechatronics* **21**, 838-850 (2015).
2. Permana, S., Grant, E., Walker, G. M. & Yoder, J. A. A review of automated microinjection systems for single cells in the embryogenesis stage. *IEEE/ASME Transactions on Mechatronics* **21**, 2391-2404 (2016).
3. Smith, G. D. & Takayama, S. Application of microfluidic technologies to human assisted reproduction. *MHR: Basic Science of Reproductive Medicine* **23**, 257-268 (2017).

4. Kashaninejad, N., Shiddiky, M. J. A. & Nguyen, N. T. Advances in microfluidics-based assisted reproductive technology: From sperm sorter to reproductive system-on-a-chip. *Advanced Biosystems* **2**, 1700197 (2018).
5. Weng, L. IVF-on-a-chip: recent advances in microfluidics technology for in vitro fertilization. *SLAS TECHNOLOGY: Translating Life Sciences Innovation* **24**, 373-385 (2019).
6. Neuman, K. C. & Nagy, A. Single-molecule force spectroscopy: optical tweezers, magnetic tweezers and atomic force microscopy. *Nature Methods* **5**, 491-505 (2008).
7. Thalhammer, G., Steiger, R., Bernet, S. & Ritsch-Marte, M. Optical macro-tweezers: trapping of highly motile micro-organisms. *Journal of Optics* **13**, 044024 (2011).
8. Dai, Y. et al. Magnetically actuated cell-robot system: precise control, manipulation, and multimode conversion. *Small* **18**, 2105414 (2022).
9. Medina-Sánchez, M., Schwarz, L., Meyer, A. K., Hebenstreit, F. & Schmidt, O. G. Cellular cargo delivery: Toward assisted fertilization by sperm-carrying micromotors. *Nano Letters* **16**, 555-561 (2016).

List of Revisions:

- ✓ We deleted the corresponding description on page 15 line 332: “*and cell manipulation*”.
- ✓ We deleted the corresponding description on page 15 line 335: “*In addition, light and magnetic controls are advantageous for in vitro manipulation of cells of different sizes. For example, when performing in vitro fertilization, it is necessary to manipulate small-sized sperm and large-sized oocyte. In this way, a single microbot can manipulate sperm, where the light field is used to control the mandibles to grab the sperm and the magnetic field is used to transport it. Subsequently, the C-shaped microbot collectives constructed by light and magnetic fields can be used to manipulate larger-sized oocyte.*”
- ✓ We deleted the corresponding description on page 15 line 338: “*Indeed, there are no pre-designed artificial gripping points in living organisms. However, the surface of tissues in living organisms often have microfold cells of various shapes, such as the stomach, small intestine, and bile ducts, which can serve as natural gripping points for*

assisting the microbot collectives to perform shape reconstruction. In addition, thanks to the flexibility of femtosecond laser two-photon polymerization to fabricate three-dimensional structures, we can also design microbots with a needle-like shape, which can penetrate and anchor in tissues. In this way, the needle-like microbot can form an artificial gripping point in vivo for the microbot collectives to accomplish shape reconfigurations and functions.”

2. Clarification on Optical Fiber Utilization: The explanation regarding the use of optical fibers in in vivo environments remains unclear. If the optical fibers can access target sites directly, the rationale for employing an additional system needs to be more convincingly detailed.

Response: We thank the reviewers for valuable comments. Our proposed strategy requires the synergistic control of a magnetic field and a light field, where the magnetic field controls the overall motion of the ant microbot and the light field controls the opening and closing of the ant microbot’s mandibles to achieve reversible assembly and disassembly among multiple units. However, the poor tissue penetration ability of the light field compared to the magnetic field limits the use of this strategy in biological organisms. Optical fibers have been used in medicine for decades because of their extreme aspect ratio profile to minimize invasiveness. As typical examples, Asma Khalid et al. coated silk fibroin on optical fibers to fabricate sensors with the ability to measure pH *in vivo*¹. Seok-Hyun Yun et al. proposed hydrogel optical fibers consisting of polymers with optical properties as the core and Ca/alginate as the cladding for continuous, real-time glucose sensing in porcine tissues². Therefore, when our proposed microbot collectives are used in situations where light is difficult to reach directly, we can introduce optical fibers as an optional tool to guide light and thus enable assembly and disassembly between multiple units.

The above-mentioned optical fibers need to be implanted in the target sites through hypodermic needle to achieve in situ detection of physiological parameters such as pH, blood glucose, and so on. Therefore, it is difficult for the fiber end surface to move

freely in target sites, which greatly affects the expansion of its function, and the direct contact between the sample and the fiber end surface might induce mechanical damage on the sample. Although researchers have proposed optical fiber tweezers, including single optical fiber tweezers and dual optical fiber tweezers to capture and manipulate tiny objects in a non-contact and damage-free manner, there are still many challenges. The single fiber tweezers can realize the optical trapping and manipulation based on a single fiber³. Since the requirement for single fiber tweezers to capture particles is the light focusing at the fiber end thereby exerting an optical gradient force on the particles near the end of the fiber, it is necessary to use a tapered optical fiber with a parabolic or convex surface. However, the inability of fiber implanted in *in vivo* environments to move freely limits its application, and this method may lead to localized heating and cause optical damage. The dual fiber tweezers mainly use the scattering force to trap particles cooperatively by two optical fibers. In addition, the dual fiber tweezers do not require focused light, resulting in minimal radiation damage to living cells⁴. However, the inability of fibers implanted in *in vivo* environments to move freely limits their application. In addition to the above respective drawbacks, they also have some disadvantages common to optical tweezers, including poor anti-interference capability. Compared to optical tweezers, our proposed coordinated control strategy of magnetic and optical fields can effectively circumvent the above problems. The microbot collectives assembled by different number of magnetic units under the coordinated control of magnetic and light fields can manipulate objects ranging from a few micrometers to several tens of micrometers, where the light field only needs to be in-suit to control the opening and closing of the mandibles of the ant microbot. This strategy is not only more resistant to interference, but also avoids suffering from localized heating and optical damage.

Although our proposed strategy avoids many of the problems that arise from the direct use of optical fibers, our strategy is currently difficult to use in *in vivo* environments due to many challenges that need to be overcome, and is more suitable for use in a lab-on-a-chip. According to the editor's suggestion, we deemphasized this part and do not

mention it in the revised manuscript and supplementary materials.

References:

1. Khalid, A. et al. Silk: A bio-derived coating for optical fiber sensing applications. *Sensors and Actuators B: Chemical* **311**, 127864 (2020).
2. Yetisen, A. K. et al. Glucose-sensitive hydrogel optical fibers functionalized with phenylboronic acid. *Advanced Materials* **29**, 15 (2017).
3. Xin, H., Xu, R. & Li, B. Optical trapping, driving and arrangement of particles using a tapered fibre probe. *Scientific reports* **2**, 818 (2012).
4. Gherardi, D. M., Carruthers, A. E., Čížmár, T., Wright, E. M. & Dholakia, K. A dual beam photonic crystal fiber trap for microscopic particles. *Applied Physics Letters* **93**, 4 (2008).

3. Experimental Validation of New Figures: The addition of Figure Fig. R1.2 re-emphasizes certain aspects of the manuscript. It is crucial that the claims and conceptual illustrations in this figure are supported by experimental evidence, particularly since they appear to play a significant role in justifying the project's motivation. This is closely related to my previous comment 4. Since Fig. 5 is still there, please do this experiment.

Response: We sincerely thank you for your comments. Currently, researchers have proposed using artificial magnetically controlled microbots to manipulate sperms. As a typical example, Oliver G. Schmidt et al. showed the capture and transportation of immobile, but otherwise functional sperms to the oocyte by coupling them to artificial helical micromotors that can be actuated by rotating magnetic fields¹. This study has demonstrated that microbots provide a new tool for in vitro fertilization, although there are many challenges in practical applications. However, limited by our existing experimental conditions, it is difficult for us to experimentally verify the process of in vitro fertilization. In addition, according to the editor's suggestion, we deemphasized the part on cell manipulation in the revised manuscript. Furthermore, we would like to reemphasize that the main contribution of this work to propose a microbot collective

reconfiguration strategy that relies on mechanical assembly and disassembly.

Reviewer #2 mentioned in previous comment 3 that the approach we propose is difficult to use for cell manipulation and is better focused on cargo transportation. According to advice from the editor and reviewer, we have deemphasized the cell manipulation. In Figure 5, we use the drug carrier as delivery object to investigate the transportation capabilities of the microbot collectives, rather than using microbot collectives for direct cell manipulation.

Reference:

1. Medina-Sánchez, M., Schwarz, L., Meyer, A. K., Hebenstreit, F. & Schmidt, O. G. Cellular cargo delivery: Toward assisted fertilization by sperm-carrying micromotors. *Nano Letters* **16**, 555-561 (2016).

List of Revisions:

- ✓ We deleted the corresponding description on page 15 line 332: *“and cell manipulation”*.
- ✓ We deleted the corresponding description on page 15 line 335: *“In addition, light and magnetic controls are advantageous for in vitro manipulation of cells of different sizes. For example, when performing in vitro fertilization, it is necessary to manipulate small-sized sperm and large-sized oocyte. In this way, a single microbot can manipulate sperm, where the light field is used to control the mandibles to grab the sperm and the magnetic field is used to transport it. Subsequently, the C-shaped microbot collectives constructed by light and magnetic fields can be used to manipulate larger-sized oocyte.”*
- ✓ We deleted the corresponding description on page 15 line 338: *“Indeed, there are no pre-designed artificial gripping points in living organisms. However, the surface of tissues in living organisms often have microfold cells of various shapes, such as the stomach, small intestine, and bile ducts, which can serve as natural gripping points for assisting the microbot collectives to perform shape reconstruction. In addition, thanks to the flexibility of femtosecond laser two-photon polymerization to fabricate three-dimensional structures, we can also design microbots with a needle-like shape, which can penetrate and anchor in tissues. In this way, the needle-like microbot can form an*

artificial gripping point in vivo for the microbot collectives to accomplish shape reconfigurations and functions.”

4. Foundation of Manuscript’s Motivation: The manuscript’s motivation, outlined in lines 56-57, appears to be based on the premise that “microbot collectives based on these intelligent units and their applications have not been validated.” This foundational statement seems rather tenuous. It would strengthen the manuscript if you could more explicitly articulate the rationale behind the chosen experimental approach and its significance in advancing the field.

Response: We thank you very much for your suggestion. According to your advice, the statement of motivation in the manuscript has been revised.

List of Revisions:

✓ We revised the corresponding description on page 3 line 56: *“Nevertheless, from the view point of the number of microbots, current researches generally focus on a single deformable microbot. Despite the fact that reversible, dynamic and fast deformation or motion of individual microbot can be realized controllably, stable and reversible connections between multiple deformable microbots to form collectives as well as their locomotion and applications have not been validated.”*

Reviewer #2 (Remarks to the Author):

Thank you for taking the time to review my comments - I especially appreciate seeing the new robot shapes. Overall, I'm satisfied with this version of the paper.

Response: We thank the reviewer for the positive assessments and constructive comments.

Reviewer #3 (Remarks to the Author):

Review of “Ant-inspired reconfigurable microbot collectives”

Second review. This is a fun contribution. The ability to lock individual components in at 0/90/180/270 degree angles to make rigid attachments using a default closed gripper is clever. The experiments are interesting and the results are easy to understand. I didn't review the supplemental material carefully. The “Tiny edits” are suggestions to make the paper easier to read.

Response: Many thanks to the reviewer for careful review and positive comments of this work. We have answered each comment and made the appropriate changes in the revised version. Please review the details of the point-by-point responses below.

“BIG ISSUES” In order of importance

1. Many images in the supplementary material are more exciting than the revised paper. I really like fig 14 and fig. 15 and fig. 16 in the supplemental material.

Response: Thank you very much for your positive comments. Supplementary Figures 14 and 15 illustrate that our assembly strategy is not limited to the shape of the microbots, which can be not only ant-shaped but also rectangular and circular. Supplementary Figure 16 illustrates that we can actuate multiple claws open at the same time by introducing a spatial light modulator to generate a multifocal light field. The above parts are the microbot shape and scale applicability extensions of our proposed methods. In addition, to maintain the consistency of the microbot shapes in the manuscript, we want to place Supplementary Figures 14-16 in the supplementary materials and hope you can understand the reasons.

“MINOR EDITS”

2. Fig. 1 a, why is the right column labeled as ‘reversible’, but the 180 degree isn't reversible. Aren't all assemblies reversible?

Response: We apologize for bringing the confusion to the reviewer. All assemblies are reversible and label has been added.

List of Revisions:

✓ We revised Figure 1a.

3. Fig. 1 b, the gap is not shown to scale. 3 ants could not bridge this large gap.

Response: We thank the reviewer for this helpful suggestion. The gap has been revised proportionally.

List of Revisions:

✓ We revised Figure 1b.

4. Fig. 2 g claims to show 1000 cycles, but only 9 cycles are shown. Was the wrong image uploaded?

Response: We apologize for bringing the confusion to the reviewer. Figure 2g has been revised.

List of Revisions:

✓ We revised Figure 2g.

5. Fig. 4 c,d,e the gap is not shown to scale. A three-ant assembly could not bridge the gap shown in fig. 4e. Please regenerate with a scale image.

Response: We thank the reviewer for pointing this out. Figure 4c-e have been revised proportionally.

List of Revisions:

✓ We revised Figure 4c-e.

6. Fig. 4 b, please add a dotted line labelled 'b_1' to show the length of one robot.

Response: We thank the reviewer for the comment. A dotted line has been added to

show the length of one microbot.

List of Revisions:

✓ We revised Figure 4b.

7. Fig. 5.d. “Contol” in legend should be “Control”.

Response: We apologize for the spelling mistake, and we have corrected it in our revised manuscript.

List of Revisions:

✓ We revised Figure 5d.

8. Fig. 5.c would benefit from a labelled color bar or legend for the green and black color.

Response: We thank the reviewer for valuable comments. In Figure 5c, the green color represents the fluorescence of living cells and the black color indicates the drug carrier. The legend has been added to Figure 5c.

List of Revisions:

✓ We revised Figure 5c.

“TINY EDITS” (e.g. typos)

9. “individual. However, various weak force-induced microbot collectives maintained by magnetic, light, and electric field still face the challenges such as unstable connections, the need for a continuous external stimuli source, and imprecise individual control. Here, we construct magnetic” -> “individual. However, various force-induced microbot collectives maintained by weak magnetic, light, or electric fields still face challenges such as unstable connections, the need for a continuous external stimuli source, and imprecise individual control. Here, we construct magnetic”.

Response: We thank the reviewer for pointing this out. The sentence has been revised.

List of Revisions:

✓ We revised the corresponding description on page 1 line 13: *“However, various force-induced microbot collectives maintained by weak magnetic, light, or electric fields still face challenges such as unstable connections, the need for a continuous external stimuli source, and imprecise individual control.”*

10. “bees¹, fishes², and birds³, causing these collectives tend to be unstable and vulnerable to disruption. Contrast, more stable collectives are formed by individuals intertwining with each other through” -> “bees¹, fishes², and birds³, causing these collectives to be unstable and vulnerable to disruption. In contrast, more stable collectives are formed by individuals intertwining with each other through”.

Response: We thank the reviewer for the comment. The sentence has been revised.

List of Revisions:

✓ We revised the corresponding description on page 2 line 30: *“In the first category there is no direct contact between individuals, as in the case of bees, fishes, and birds, causing these collectives to be unstable and vulnerable to disruption. In contrast, more stable collectives are formed by individuals intertwining with each other through mouth or limb deformations as observed in ants, or by one biting on the base of the tail of another in the case of shrews.”*

11. “in the case of shrews⁶. These collectives are formed by individuals locking onto each other can” -> “in the case of shrews⁶. These collectives are formed by individuals locking onto each other. They can”

Response: We thank the reviewer for this suggestion. The sentence has been revised.

List of Revisions:

✓ We revised the corresponding description on page 2 line 33: *“These collectives are formed by individuals locking onto each other. They can resist interference and always remain as an integrated whole.”*

This one is more confusing. Perhaps:

12. “typical examples, Sitti et al. created cubic liquid crystal elastomer voxels with predefined director field orientations and stitched these voxels into lines, grids, and skeletal structures for” -> “typical examples, Sitti et al. created cubic liquid crystal elastomer voxels with predefined field orientations directions and stitched these voxels into lines, grids, and skeletal structures for”.

Response: We thank the reviewer for pointing this out. The sentence has been revised.

List of Revisions:

✓ We revised the corresponding description on page 3 line 50: *“As typical examples, Sitti et al. created cubic liquid crystal elastomer voxels with predefined field orientations directions and stitched these voxels into lines, grids, and skeletal structures for programmable thermally-triggered morphology changes in three dimensions.”*

13. “process³²⁻³⁶, and then the silver nanoparticles (Ag NPs) are deposited onto the head of the ant” -> “process³²⁻³⁶, and then silver nanoparticles (Ag NPs) are deposited onto the head of the ant”.

Response: We thank the reviewer for the comment. The sentence has been revised.

List of Revisions:

✓ We revised the corresponding description on page 3 line 63: *“The rigid body and flexible joints of each ant microbot are fabricated by sequential polymerization of magnetic photoresist and thermal stimuli-responsive hydrogel by two-photon polymerization (TPP) process, and then silver nanoparticles (Ag NPs) are deposited onto the head of the ant microbot based on photoreduction.”*

14. “Ag NPs, the ant mandibles could open immediately when the laser is focused on the ant head.” -> “Ag NPs, the ant mandibles open immediately when the laser is

focused on the ant head.”

Response: We thank the reviewer for this suggestion. The sentence has been revised.

List of Revisions:

✓ We revised the corresponding description on page 3 line 67: *“Due to the powerful photothermal conversion effect of the Ag NPs, the ant mandibles open immediately when the laser is focused on the ant head.”*

15. 109 & 133: “In order to” -> “To”.

Response: We thank the reviewer for pointing this out. The sentence has been revised.

List of Revisions:

✓ We revised the corresponding description on page 5 line 113: *“In this work, to quantitatively investigate the expansion and contraction properties of this hydrogel, micro circular plates with a diameter of 20 μm and a thickness of 3 μm are fabricated and measured and full contraction in 8 ms is observed when the microplate is illuminated.”*

✓ We revised the corresponding description on page 6 line 137: *“To obtain insight into the degree of bending deformation caused by different SRTs, theoretical simulation is conducted to verify the bending shape of the joints, and the simulated bending direction and angle match well with the experimental results, indicating the feasibility of this design strategy (Figure 2e).”*

16. 125 “It should be noted that the” -> “The”.

Response: We thank the reviewer for the comment. The sentence has been revised.

List of Revisions:

✓ We revised the corresponding description on page 6 line 129: *“The hydrogel joint consists of two layers with different SRT, which result in distinct degrees of crosslinking, and consequently, the hydrogel joints can exhibit asymmetric deformation.”*

17. 184 “the latter mandibles could encircle” -> “the latter mandibles encircle”.

Response: We thank the reviewer for this suggestion. The sentence has been revised.

List of Revisions:

✓ We revised the corresponding description on page 9 line 187: *“Afterward, the latter continues to move forward driven by the magnetic field so that the latter mandibles encircle the former tail.”*

18. 195 “into the J shape” -> “into a J shape”.

Response: We thank the reviewer for pointing this out. The sentence has been revised.

List of Revisions:

✓ We revised the corresponding description on page 9 line 198: *“Benefiting from the controlled and selective assembly, these independent units are reassembled into a J shape.”*

19. 238 “the stability of C shape is larger than that of I shape” -> “the stability of the C shape is larger than that of the I shape”.

Response: We thank the reviewer for the comment. The sentence has been revised.

List of Revisions:

✓ We revised the corresponding description on page 11 line 240: *“It can be seen that the amplitude needed to disperse the assembled architecture gradually decreases as the applied frequency increases for both assembly methods and the stability of the C shape is larger than that of the I shape.”*

20. “The ant colonies in nature contain many individuals, which can collaborate to accomplish various tasks. In contrast, the number of individuals in the collectives formed by our proposed assembly strategy is difficult to be as large as that of ant

colonies.” -> “Ant colonies in nature contain many individuals, which can collaborate to accomplish various tasks. In contrast, it is difficult to use our assembly strategy for large numbers of individuals.”

Response: We thank the reviewer for this suggestion. The sentence has been revised.

List of Revisions:

✓ We revised the corresponding description on page 12 line 273: “*Ant colonies in nature contain many individuals, which can collaborate to accomplish various tasks. In contrast, it is difficult to use our assembly strategy for large numbers of individuals.*”

21. 277 “allow only I-shaped” -> “allow only an I-shaped”.

Response: We thank the reviewer for pointing this out. The sentence has been revised.

List of Revisions:

✓ We revised the corresponding description on page 13 line 280: “*The width of the entrance and exit of the maze allow only an I-shaped three-unit microbot to pass through and the entrance and exit of the maze have a gripping point to facilitate the switching of the assembly pattern.*”

22. 309 “Figure 5b is the time-lapse images of drug carrier delivery” -> “Figure 5b shows time-lapse images of drug delivery”.

Response: We thank the reviewer for the comment. The sentence has been revised.

List of Revisions:

✓ We revised the corresponding description on page 14 line 312: “*Figure 5b shows time-lapse images of drug carrier delivery, and the images from each location have been stitched together to demonstrate the full path.*”

23. 311 “increases over time and need” -> “increases over time and needs”.

Response: We thank the reviewer for this suggestion. The sentence has been revised.

List of Revisions:

✓ We revised the corresponding description on page 14 line 313: *“After the carrier is delivered to the targeted position, the extent and intensity of the red fluorescence gradually increases over time and needs approximately 20 min to fill the entire area of the experimental group, indicating that the drugs had diffused rapidly from the carrier (Supplementary Figure 22).”*

24. 321 “Microbot collectives have a superior ability to accomplish complex tasks compared to individual microbot.” -> “Microbot collectives have a superior ability to accomplish complex tasks compared to individual microbots.”

Response: We thank the reviewer for pointing this out. The sentence has been revised.

List of Revisions:

✓ We revised the corresponding description on page 14 line 323: *“Microbot collectives have a superior ability to accomplish complex tasks compared to individual microbots.”*

25. 325 “the ability to traverse gap” -> “the ability to traverse gaps”.

Response: We thank the reviewer for the comment. The sentence has been revised.

List of Revisions:

✓ We revised the corresponding description on page 15 line 327: *“The ant microbot collectives based on mechanically deformed assemblies designed in this study demonstrate assembly morphological diversity and strong stability and the ability to traverse gaps and environmental adaptability and portability under different assembly morphologies, which broaden the behaviors and functions of microbot collectives.”*

26. 339 “In our proposed assembly strategy, gripping point” -> “In our proposed assembly strategy, a gripping point”.

Response: We thank the reviewer for this suggestion. The sentence has been revised.

List of Revisions:

- ✓ We revised the corresponding description on page 15 line 335: *“In our proposed assembly strategy, a gripping point is required to assist in the transformation of the assembly shape.”*

REVIEWERS' COMMENTS

Reviewer #1 (Remarks to the Author):

I have to thank the reviewer for answering my questions. Here are my replies:

Question 1: Optical Tweezers and Light-Driven Micro-Manipulation

For question one, I have to point out that there are many optical tweezers-driven, or simply light-driven mechanical structures that can be used to manipulate cells. For these approaches, the light does not illuminate the cells directly. The light manipulation community has known this for a very long time, making the concept proposed here not new. Moreover, table R1 shows this work does not damage the cell. However, when the force is not well controlled, it always has adverse effects, in addition to other effects caused by the contact of the cell with the end-effector, which is the 3D-printed structures here. Therefore, it is more appropriate to state that the proposed method is "unverified" in this table because no further data is provided to support that this proposed method causes no damage (or I missed them?).

Question 2: In Vivo Applications and Combining Light and Magnetic Actuation

Thanks for being honest by stating that "our strategy is currently difficult to use in in vivo environment" after reviewing so much optical fiber literature. But unfortunately, it verifies my guess that the in vivo application is not possible for the proposed method, in addition to the challenge in question one regarding in vitro application.

Frankly, the combination of light and magnetic actuation in this case, while seemingly beneficial in increasing the control degrees of freedom (DOF) by allowing independent manipulation of motion and the gripper, also compromises the inherent advantages of each method. Magnetic actuation's wireless control and light actuation's precise steering capabilities, high energy density (e.g., laser), and simple manipulation (objects usually follow the light spot in several mechanisms) are all undermined when combined, potentially reducing the overall efficacy of the approach.

Question 3: Manuscript's Contribution on Cell Manipulation

By answering this question, the manuscript has decided to shift its focus away from cell manipulation, which impacts its overall contribution. Given the extensive body of work in the field of miniature robots, numerous established methods already exist for moving cargos. Many of these methods can perform similar tasks effectively. Therefore, the novelty and significance of the proposed technique must be reconsidered in light of these existing technologies.

Question 4: Motivation and Research Necessity

Although there are no significant questions regarding the fourth aspect, it remains beneficial to delve into the background and motivation for this research. A thorough understanding of why this study is necessary can enhance its relevance and impact, ensuring that it addresses critical gaps or unmet needs in the field.